# The Generalized Skew Spectrum of Graphs

**Armando Bellante** [1 2 3]   **Martin Plávala** [4 5]   **Alessandro Luongo** [6 7]

## Abstract

This paper proposes a family of permutation-invariant graph embeddings, generalizing the Skew Spectrum of graphs of Kondor & Borgwardt (2008). Grounded in group theory and harmonic analysis, our method introduces a new class of graph invariants that are isomorphism-invariant and capable of embedding richer graph structures - including attributed graphs, multilayer graphs, and hypergraphs - which the Skew Spectrum could not handle. Our generalization further defines a family of functions that enables a trade-off between computational complexity and expressivity. By applying generalization-preserving heuristics to this family, we improve the Skew Spectrum's expressivity at the same computational cost. We formally prove the invariance of our generalization, demonstrate its improved expressiveness through experiments, and discuss its efficient computation.

## 1. Introduction

Graph-structured data is ubiquitous across domains, from molecular structures in chemistry (Kosmala et al., 2023) to social networks in computational social science (Nettleton, 2013), or fraud detection in banking (Pourhabibi et al., 2020). However, developing adequate graphs representations remains a fundamental challenge in graph analysis (Hamilton, 2020). Traditional graph representations,

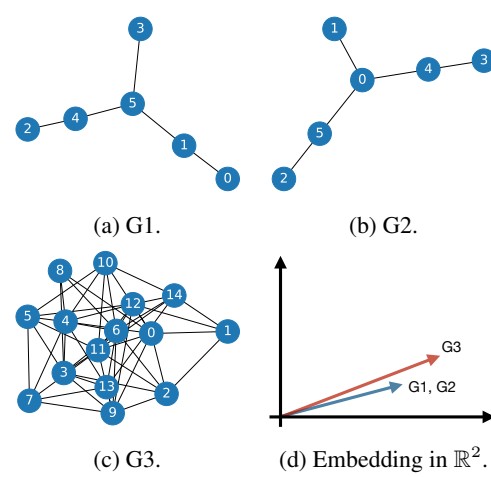

(a) G1.   (b) G2.

(c) G3.   (d) Embedding in $\mathbb{R}^2$.

*Figure 1.* Three graphs and a permutation-invariant embedding.

such as adjacency matrices, are highly sensitive to the conventional node numbering used when acquiring the graph data. For instance, graphs $G1$ and $G2$ in Figure 1 are isomorphic, but their different node numbering would yield distinct adjacency matrices. Graph representations that are sensitive to initial node numbering make learning on graphs a harder task, similarly to how rotation- and translation-sensitive representations make learning on images challenging. This motivates the need for *permutation-invariant graph embeddings*, also known as *graph invariants*.

A permutation-invariant embedding is a mapping that represents graphs as vectors while ensuring that isomorphic graphs are assigned the same vector. The *expressivity*, or *completeness*, of such an embedding is determined by its ability to distinguish non-isomorphic graphs. An embedding is *complete* if it maps all non-isomorphic graphs to different vectors. However, as of today, achieving both computational efficiency and completeness requires a trade-off, as determining wether two graphs are isomorphic is a paramount example of an NP-intermediate problem (Babai, 2016).

Various approaches have been proposed to construct graph invariants, and the challenge has gained renewed importance with the rise of graph neural networks (GNNs) (Scarselli et al., 2008) and their applications across scientific disciplines (Corso et al., 2024). Modern methods primarily rely on message-passing neural networks (Gilmer et al., 2017)

[1]Max-Planck-Institut für Quantenoptik, Hans-Kopfermann-Str. 1, 85748 Garching, Germany [2]Munich Center for Quantum Science and Technology (MCQST), Schellingstr. 4, 80799 München, Germany [3]Dipartimento di Elettronica, Informazione e Bioingegneria, Politecnico di Milano, Milan, Italy [4]Institute of Theoretical Physics, Leibiniz Universität Hannover, Appelstraße 2, 30167 Hannover, Germany [5]Naturwissenschaftlich-Technische Fakultät, Universität Siegen, Siegen, Germany [6]Centre for Quantum Technologies, National University of Singapore, Singapore [7]Inveriant Pte. Ltd., Singapore. Correspondence to: Armando Bellante <armando.bellante@polimi.it>.

*Proceedings of the $42^{nd}$ International Conference on Machine Learning*, Vancouver, Canada. PMLR 267, 2025. Copyright 2025 by the author(s).

or spectral properties of the graph Laplacian (Hammond et al., 2011). While these approaches have achieved practical success, they often lack theoretical guarantees about their expressivity beyond the Weisfeiler-Lehman test (Morris et al., 2019; Xu et al., 2019) or do not generalize naturally to richer graph structures like attributed graphs, multilayer graphs, and hypergraphs. This limitation becomes particularly crucial as such complex graph structures increasingly appear in real-world applications, from multi-omics biological networks to heterogeneous social networks.

This leads us to investigate the following research questions:

*How can we construct permutation-invariant graph embeddings that (i) handle rich graph structures beyond simple graphs, (ii) are mathematically well-grounded, and (iii) offer practical computational efficiency?*

*Furthermore, can we develop a theoretical framework that enables principled trade-offs between computational efficiency and completeness?*

Group theory offers a principled and complementary framework for constructing graph invariants with provable properties (Bronstein et al., 2021; Dehmamy et al., 2021; Batatia et al., 2023). A seminal contribution in this direction was the Skew Spectrum (Kondor & Borgwardt, 2008), which introduced a rigorous method for constructing graph invariants through functions on the symmetric group. While theoretically elegant, the original Skew Spectrum could not handle rich graph structures and offered no clear pathway to trade off between computational complexity and expressivity. Our work revisits and improves these older group-theoretic foundations to address modern challenges in graph representation, opening new research directions.

In this work, we introduce a generalization of the Skew Spectrum. Our contributions are threefold:

1. *Multi-Orbit Spectra*: We develop a mathematically rigorous family of permutation-invariant graph embeddings that naturally extend to attributed graphs, multilayer graphs, and hypergraphs;
2. *$k$-Correlation Spectra*: We establish a theoretical framework that enables principled trade-offs between computational complexity and expressivity;
3. *Doubly-Reduced $k$-Spectra*: We introduce heuristics that preserve the generalizations and improve the expressivity of the Skew Spectrum without increasing its computational cost, bridging theory and practice.

We illustrate our theoretical contributions with numerical experiments, demonstrating that our generalizations significantly improve the Skew Spectrum expressivity: distinguishing richer graphs, and distinguishing more non-isomorphic simple graphs at the same computational complexity.

## 2. Background

Our work extends the seminal paper of Kondor & Borgwardt (2008) on the Skew Spectrum of graphs. In this section, we review the key concepts, laying the ground for our results.

**Notation.** The symbol $\bigoplus$ denotes a direct sum. For example, $\bigoplus_{i=1}^{n} i$ denotes a diagonal matrix with entries $1, \ldots, n$ along the diagonal. The symbol $\bigotimes$ denotes a sequence of tensor products. For instance, $\bigotimes_{i=1}^{2} v_i = v_1 \otimes v_2$. We use $[n]$ to denote the set of integers $\{1, \ldots, n\}$. Similarly, $[a, b]$ refers to the set of integers between $a$ and $b$, inclusive.

### 2.1. The Symmetric Group

The symmetric group $\mathbb{S}_n$ consists of all permutations of $n$ elements. A permutation $\sigma \in \mathbb{S}_n$ is a bijection of the indices of these elements, where $\sigma(i)$ denotes the image of index $i$ under $\sigma$. We use cycle notation, e.g., $\sigma = (1, 4, 3)$ maps $1 \rightarrow 4, 4 \rightarrow 3$, and $3 \rightarrow 1$. For $k < n$, the subgroup $\mathbb{S}_k$ a naturally embeds in $\mathbb{S}_n$, by acting on the first $k$ indices while stabilizing the last $n - k$. This induces a natural partition of $\mathbb{S}_n$ into right and left cosets, leading to the coset tranversals $\mathbb{S}_k \backslash \mathbb{S}_n$ and $\mathbb{S}_n / \mathbb{S}_k$. These right (left) sets contain exactly one representative element per coset: a unique $x \in \mathbb{S}_k \backslash \mathbb{S}_n$ ($y \in \mathbb{S}_n / \mathbb{S}_k$), such that any $\sigma \in \mathbb{S}_n$ can be decomposed as $\sigma = h_1 x$ ($\sigma = y h_2$) for some $h_1 \in \mathbb{S}_k$ ($h_2 \in \mathbb{S}_k$). This definition extends to double-coset transversals $\mathbb{S}_k \backslash \mathbb{S}_n / \mathbb{S}_k$, where any $\sigma \in \mathbb{S}_n$ can be written as $\sigma = h_3 z h_4$ for a unique $z \in \mathbb{S}_k \backslash \mathbb{S}_n / \mathbb{S}_k$ and $h_3, h_4 \in \mathbb{S}_k$. For example, $\mathbb{S}_{n-2} \backslash \mathbb{S}_n / \mathbb{S}_{n-2}$ consists of the following 7 group elements: $\{(), (n - 1, n), (n - 2, n - 1), (n - 2, n), (n - 2, n - 1, n), (n - 2, n, n - 1), (n - 3, n - 1)(n - 2, n)\}$.

We denote irreducible representations (irreps) of $\mathbb{S}_n$ by $\rho$. These are mappings from group elements to unitary matrices, and can be labeled by partitions of $n$. Specifically, we use the Young Orthogonal Representation (YOR), which has real-valued entries. The set of YORs is denoted by $\mathrm{Irr}(\mathbb{S}_n)$, and we define $\Lambda_n$ as the subset of four irreps indexed by the partitions $(n), (n - 1, 1), (n - 2, 2), (n - 2, 1, 1)$.

We refer the interested reader to Appendix A and the background sections (2.6 and 2.7) of Bellante (2024) for more.

### 2.2. Graph isomorphism and functions on $\mathbb{S}_n$

The adjacency matrix $A$ of a graph encodes its structure, with $A_{i,j}$ specifying the relationship between nodes $i$ and $j$. The construction of $A$ inherently depends on the node numbering, and since graphs are often labeled arbitrarily, the same structure can be represented by multiple distinct adjacency matrices, complicating graph comparison tasks.

Graph isomorphism is the problem of determining whether two graphs $G1$ and $G2$ have the same structure despite

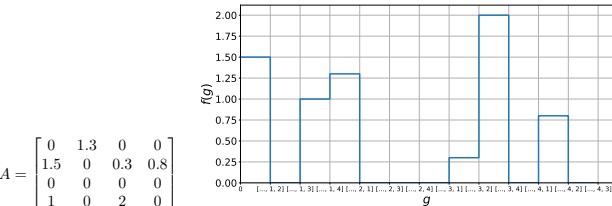

(a) Adjacency matrix.

(b) Function $f(g) = A_{g(n),g(n-1)}$. $g$ is represented as a vector where $g_i = g(i)$.

*Figure 2.* A weighted, directed graph as a function on $\mathbb{S}_4$.

differences in node numbering. Formally, for adjacency matrices $A^{(1)}$ and $A^{(2)}$, $G1$ and $G2$ are isomorphic if there exists a permutation $\sigma \in \mathbb{S}_n$ such that $A_{i,j}^{(1)} = A_{\sigma(i),\sigma(j)}^{(2)}$. Here, $\sigma$ reorders the node indices of $G2$ to align with $G1$, preserving the edge relationships.

To encode adjacency information in a manner that reflects this relationship, we define a function $f : \mathbb{S}_n \to \mathbb{R}$:

$$f(g) = A_{g(n),g(n-1)} \text{ for all } g \in \mathbb{S}_n. \tag{1}$$

This encoding captures information about node adjacency while naturally reflecting the action of permutations on node indices. Indeed, using this formalism, two graphs are isomorphic if and only if their corresponding functions satisfy:

$$\exists\, \sigma \text{ s.t. } f^{(A_1)}(g) = f^{(A_2)}(\sigma g) \text{ for all } g \in \mathbb{S}_n. \tag{2}$$

In other words, *graph isomorphism* reduces to verifying whether two functions are related by a *translation* $\sigma \in \mathbb{S}_n$. The Skew Spectrum leverages this formulation to construct graph invariants through translation-invariant functions on $\mathbb{S}_n$, using correlation functions and their Fourier transforms.

To conclude, note that graph functions (1) are sparse: they depend only on the image of the last two indices under $g$ ($g(n-1)$ and $g(n)$). More formally, $f(\sigma h) = f(\sigma)$ for all $h \in \mathbb{S}_{n-2}$. Thus, $f$ is supported on the coset transversal $\mathbb{S}_n / \mathbb{S}_{n-2}$. As shown in Fig. 2, this sparsity implies that $f$ takes distinct values on at most $n^2$ elements, rather than $n!$.

### 2.3. Fourier transform on $\mathbb{S}_n$

The Fourier transform of a function over $\mathbb{S}_n$ is defined as

$$\hat{f}(\rho) = \frac{1}{|\mathbb{S}_n|} \sum_{g \in \mathbb{S}_n} f(g)\rho(g) \tag{3}$$

where $\rho$ is an irreducible representation of $\mathbb{S}_n$.

Standard FFT algorithms require $O(n!\text{polylog}(n!))$ operations to evaluate $\hat{f}(\rho)$ for all irreps (Maslen, 1998). However, for graph-derived functions, taking values on $\mathbb{S}_n / \mathbb{S}_{n-2}$, the complexity reduces to $O(n^3)$ (or even $O(n^2)$) operations (Kondor & Borgwardt, 2008; Clausen & Hühne, 2017).

A key factor in this reduction is the projection $\tau_{\mathbb{S}_{n-2}}(\rho)$, which maps $\rho$ onto the trivial irrep of $\mathbb{S}_{n-2}$. This projection induces a sparsity structure in the Fourier transform, enabling efficient computation. This sparsity also plays a critical role in the efficient computation of the graph invariants. We formalize this sparsity in the following lemma.

**Lemma 2.1** (Fourier sparsity). *Let $f : \mathbb{S}_n / \mathbb{S}_{n-2} \to \mathbb{C}$. Its Fourier transform using YOR is a collection of sparse square matrices indexed by specific partitions of $n$. The sparsity pattern is independent of $f$:*

1. *$(n)$; size 1, one scalar entry.*
2. *$(n-1)$; size $n-1$, 2 non-zero columns.*
3. *$(n-2,2)$; size $\frac{n(n-3)}{2}$, 1 non-zero columns.*
4. *$(n-2,1,1)$; size $\frac{(n-1)(n-2)}{2}$, 1 non-zero columns.*

*We use $\Lambda_n$ to denote this set of partitions.*

For a detailed discussion on $\tau_{\mathbb{S}_{n-2}}$ and its connection to the sparsity pattern, see Appendix B.

### 2.4. The Skew Spectrum of Graphs

Kondor & Borgwardt (2008) introduced a *translation-invariant* function on $\mathbb{S}_n$ as the spectrum (the Fourier transform) of a *triple correlation* function, which they termed the Skew Spectrum.

**Definition 2.2** (Skew Spectrum). Let $f : \mathbb{S}_n \to \mathbb{C}$. An entry of the Skew Spectrum is defined as

$$\hat{\mathcal{S}}_f(g_1, \rho) = \sum_{\substack{\tilde{g} \in \mathbb{S}_n \\ \tilde{g}_2 \in \mathbb{S}_n}} \frac{f(\tilde{g})f(\tilde{g}g_1)f(\tilde{g}_2)}{|\mathbb{S}_n|^2} \rho(\tilde{g})^{\dagger}\rho(\tilde{g}_2). \tag{4}$$

A Skew Spectrum entry is a (sparse) matrix. The Skew Spectrum is uniquely determined by a small subset of entries - specifically, it is indexed by 7 group elements ($\mathbb{S}_{n-2}\backslash\mathbb{S}_n/\mathbb{S}_{n-2}$) and 4 irreps ($\Lambda_n$). Moreover, (4) can be expressed as $\hat{\mathcal{S}}_f(g_1, \rho) = \hat{r}(g_1, \rho)^{\dagger}\hat{f}(\rho)$, where $\hat{r}(g_1, \rho) = \sum_{\tilde{g}\in\mathbb{S}_n} \frac{f(\tilde{g})f(\tilde{g}g_1)}{|\mathbb{S}_n|}\rho(\tilde{g})$ is the Fourier transform of another function $r(g_1, \sigma)$. This decomposition, Fourier transforms, and sparsity are key to *efficient computation*.

The Skew Spectrum has up to 85 non-zero scalar values (features) across all the entries, and they can be computed in $O(n^6)$ operations. To improve efficiency, the authors proposed a way to trade expressivity for computational cost, introducing a reduced invariant.

**Definition 2.3** (Reduced Skew Spectrum). Let $f : \mathbb{S}_n \to \mathbb{C}$. An entry of the reduced Skew Spectrum is defined as

$$\hat{\mathcal{S}}_f(g_1, \rho) = \sum_{\substack{\tilde{g} \in \mathbb{S}_n \\ \tilde{g}_2 \in \mathbb{S}_n}} \frac{f(\tilde{g})f(\tilde{g}g_1)f(\tilde{g}_2)}{|\mathbb{S}_n|^2} \rho(\tilde{g})^{\dagger}\tau_{\mathbb{S}_{n-2}}(\rho)^{\dagger}\rho(\tilde{g}_2).$$

where $\tau$ projects $\rho$ onto the trivial representation of $\mathbb{S}_{n-2}$.

This projection reduces the domain of $r(g_1, \sigma)$ to $\mathbb{S}_n/\mathbb{S}_{n-2}$, ensuring the same sparsity pattern as in $\hat{f}(\rho)$ and further simplifying the computation. The reduced Skew Spectrum's efficiency is summarized in the following result.

**Theorem 2.4** (Reduced Skew Spectrum computation). *Let* $f : \mathbb{S}_n/\mathbb{S}_{n-2} \to \mathbb{C}$. *We can compute its reduced Skew Spectrum in* $O(n^3)$ *operations. The reduced Skew Spectrum has up to* 49 *non-zero entries.*

The reduced Skew Spectrum is the key contribution of Kondor & Borgwardt (2008), validated experimentally as an effective and efficient tool for representing graphs. In the next sections, we will present our generalization.

## 3. Richer graph structures and multiple orbits

While the Skew Spectrum is a powerful invariant for weighted and directed graphs, it does not fully capture richer graph structures such as graphs with node or edge features, multilayer graphs, or hypergraphs. To address these limitations, we introduce the Multi-Orbit Skew Spectrum, enabling the analysis of these complex structures.

Richer graph structures are typically represented by a collection of data structures (e.g., adjacency matrices, node feature vectors, etc.) indexed by the graph nodes. Under node renumbering, the elements within these structures permute independently without mixing (e.g., node features remain distinct from edge weights). These distinct subsets of data are referred to as *orbits*, and two such graph structures are isomorphic if there exists a single permutation that aligns all their orbits.

To illustrate, consider graphs $G1$ and $G2$ in Figure 1. Treating node numbers as labels, the two graphs are not isomorphic; e.g., node 5 has a high degree in $G1$ but a lower degree in $G2$. For isomorphism, both the adjacency matrix and the node labels must align under the same permutation.

A natural extension of the invariant is to define separate functions for each orbit (e.g., one for the adjacency matrix and one for node labels) and concatenate their respective Skew Spectra. However, this approach fails to account for correlations between orbits. For example, the individual spectra of $G1$ and $G2$ would coincide, even though the permutations linking the adjacency matrix and node labels differ. This information loss limits the invariant's completeness.

To overcome this, we replace scalar functions with vector-valued functions, enabling a unified representation of multiple orbits. Specifically, we define a function $f : \mathbb{S}_n/\mathbb{S}_{n-2} \to \mathbb{C}^d$ as the direct sum of single-orbit functions:

$$f(g) = \bigoplus_{i \in [d]} f_i(g) \qquad (5)$$

The pointwise product required for the Skew Spectrum is then replaced by the tensor product.

**Definition 3.1** (Multi-Orbit Skew Spectrum). Let $f : \mathbb{S}_n \to \mathbb{C}^d$. A Multi-Orbit Skew Spectrum entry is defined as

$$\hat{\mathcal{S}}_f(g_1, \rho) = \sum_{\substack{\tilde{g} \in \mathbb{S}_n \\ \tilde{g}_2 \in \mathbb{S}_n}} \frac{f(\tilde{g}) \otimes f(\tilde{g}g_1) \otimes f(\tilde{g}_2)}{|\mathbb{S}_n|^2} \otimes \left(\rho(\tilde{g})^\dagger \rho(\tilde{g}_2)\right).$$

Expanding the direct sums in $f(g)$, we obtain:

$$\hat{\mathcal{S}}_f(g_1, \rho) = \bigoplus_{(i_1,i_2,i_3) \in [d]^3} \hat{r}_{(i_1,i_2)}(g_1, \rho)^\dagger \hat{f}_{i_3}(\rho) \qquad (6)$$

where $\hat{r}_{(i_1,i_2)}(g_1, \rho) = \sum_{\tilde{g} \in \mathbb{S}_n} \frac{f_{i_1}(\tilde{g})f_{i_2}(\tilde{g}g_1)}{|\mathbb{S}_n|}\rho(\tilde{g})$. This formulation introduces terms $f_i(\tilde{g})f_j(\tilde{g}g_1)$, for $i \neq j$, which capture correlations between orbits and enhance the spectrum's ability to distinguish between graphs. Our intuition comes from a quantum theory perspective, where these terms can be viewed as interference terms. In quantum theory interference terms capture superposition-related phenomena where the whole can be greater than sum of the parts. Similarly, the Multi-Orbit Skew Spectrum is more expressive than the concatenation of individual spectra.

We formally prove the invariance of the Multi-Orbit Skew Spectrum in Section 4, where it emerges as a special case of the broader framework we propose. Its reduced version is akin to the one of the original Skew Spectrum: each term of the direct sum in (6) is computed as a reduced Skew Spectrum (Def. 2.3). Its computation is detailed in Section 6 and has the following complexity.

**Theorem 3.2** (Reduced Multi-Orbit Skew Spectrum). *Let* $f : \mathbb{S}_n/\mathbb{S}_{n-2} \to \mathbb{C}^d$. *Computing its reduced Multi-Orbit Skew Spectrum takes* $O(d^2n^3 + d^3n^2)$ *operations. The resulting embedding has up to* $49d^3$ *non-zero entries.*

In the remainder of this section, we demonstrate how to construct Multi-Orbit functions for various graph structures. These examples illustrate the spectrum's versatility in capturing complex data representations.

**Self-loops.** Graphs with self-loops naturally split into two orbits: diagonal and off-diagonal entries, which do not mix under permutations. Define $f_1 = A_{\sigma(n),\sigma(n-1)}$ for off-diagonal entries and $f_2 = A_{\sigma(n),\sigma(n)}$ for diagonal entries.

**Node features.** If each node has a vector $x_i$ of $p$ features, the Multi-Orbit function $f$ spans $d = p + 1$ orbits. Specifically, $f_i(\sigma) = x_{i,\sigma(n)}$ for $i \in [p]$ capturing the features, and $f_d(\sigma) = A_{\sigma(n),\sigma(n-1)}$, encoding the adjacency.

**Edge features.** When each edge has up to $d$ features, the graph can be represented as a tensor $A \in \mathbb{C}^{d \times n \times n}$, where $A_{l,i,j}$ encodes the $l^{\text{th}}$ feature of the edge between nodes $i$ and $j$. The function $f_i(\sigma) = A_{i,\sigma(n),\sigma(n-1)}$ captures these features, with the number of orbits scaling with the features.

**Multilayer graphs.** Graphs with multiple edge layers, each representing distinct relationships (e.g., friendship or work connections in social networks), can be represented using separate adjacency matrices. For layer $i$, define $f_i(\sigma) = A_{i,\sigma(n),\sigma(n-1)}$, where $i$ indexes the layers.

**Hypergraphs.** In hypergraphs, edges connect multiple nodes. A simple representation uses $A \in \mathbb{C}^{d \times n}$, with rows corresponds to edges and $f_i(\sigma) = A_{i,\sigma(n)}$ specifies node-edge relationships. Although $f$ is defined over $\mathbb{S}_n/\mathbb{S}_{n-1}$, potentially reducing the computation, the number of orbits matches the number of edges, which can be large.

Alternatively, for hypergraphs with edges connecting up to $q$ nodes, an adjacency tensor $A_{i_1,\dots,i_q}$ can encode whether an edge exists between nodes $i_1, \dots, i_q$. A fictitious node $n + 1$ handles edges with fewer than $q$ nodes. This defines $f(\sigma) = A_{\sigma(n),\dots,\sigma(n-(q-1))}$ on $\mathbb{S}_n/\mathbb{S}_{n-q}$. While beyond the scope of this paper, Clausen & Hühne (2017) provide efficient Fourier transforms on $\mathbb{S}_n/\mathbb{S}_{n-q}$, which could be used to extend our invariant.

## 4. Spectra of higher correlations

The Skew Spectrum and our Multi-Orbit generalization are Fourier transforms over the last function entry of a triple correlation $\mathcal{S}_f^{(3)}(\sigma_1, \sigma_2) = \sum_{\tilde{\sigma} \in \mathbb{S}_n} \frac{f(\tilde{\sigma})f(\tilde{\sigma}\sigma_1)f(\tilde{\sigma}\sigma_2)}{|\mathbb{S}_n|}$.

In signal analysis, it is well-known that correlation functions constitute translation invariants, because summing over the entire domain effectively averages out translations. Our idea for the second generalization is to compute higher correlations, aiming to increase the completeness of the invariant at the expense of higher computation complexity. Formally, we define a (Multi-Orbit) $k$-correlation as follows.

**Definition 4.1** ($k$-correlation). Let $f : \mathbb{S}_n \to \mathbb{C}^d$. An entry of the $k$-correlation function is defined as

$$\mathcal{S}_f^{(k)}(G_{k-1}) = \frac{1}{|\mathbb{S}_n|} \sum_{\tilde{g} \in \mathbb{S}_n} f(\tilde{g}) \bigotimes_{l=1}^{k-1} f(\tilde{g}g_l) \qquad (7)$$

where $G_{k-1} = (g_1, \dots, g_{k-1}) \in (\mathbb{S}_n)^{k-1}$.

Our family of invariants - the $k$-Spectrum entries - is given by Fourier transforms over the last entry of a $k$-correlation function. This establishes a bijection with the correlation functions, while helping the computation.

**Definition 4.2** ($k$-Spectrum). Let $f : \mathbb{S}_n \to \mathbb{C}^d$. An entry of the $k$-Spectrum is defined as

$$\hat{\mathcal{S}}_f^{(k)}(G_{k-2}, \rho) = \bigoplus_{I_k} \hat{r}_{I_{k-1}}^{(k)}(G_{k-2}, \rho)^{\dagger} \hat{f}_{i_k}(\rho). \qquad (8)$$

Here $I_k = (i_1, \dots, i_k) \in [d]^k$ is a list of indices, and

$$\hat{r}_{I_{k-1}}^{(k)}(G_{k-2}, \rho) = \sum_{\tilde{g} \in \mathbb{S}_n} \frac{f_{i_{k-1}}(\tilde{g}) \prod_{l=1}^{k-2} f_{i_l}(\tilde{g}g_l)}{|\mathbb{S}_n|} \rho(\tilde{g}). \qquad (9)$$

The invariance of the $k$-Spectrum follows directly from the one of the $k$-correlation functions, holding for both the Skew Spectrum ($k = 3$) and our Multi-Orbit extension.

**Theorem 4.3** (Invariance). *Let $f_1, f_2 : \mathbb{S}_n \to \mathbb{C}^d$ be equivalent to up to left-translation, such that $f_1(\sigma) = f_2(\overline{\sigma}\sigma)$ for some fixed $\overline{\sigma} \in \mathbb{S}_n$ and every $\sigma \in \mathbb{S}_n$. Then, the Multi-Orbit $k$-Spectra of $f_1$ and $f_2$ are identical. In other words, for each $G_{k-2} \in (\mathbb{S}_n)^{k-2}$, $\rho \in \mathrm{Irr}(\mathbb{S}_n)$, we have $\hat{\mathcal{S}}_{f_1}^{(k)}(G_{k-2}, \rho) = \hat{\mathcal{S}}_{f_2}^{(k)}(G_{k-2}, \rho)$.*

*Proof.* Because of the bijection, we show the translation invariance of $k$-correlations. By substituting $g' = \overline{\sigma}\tilde{g}$, we obtain $\mathcal{S}_{f_1}^{(k)}(G_{k-1}) = \sum_{\tilde{g} \in \mathbb{S}_n} \frac{f_2(\overline{\sigma}\tilde{g}) \bigotimes_{l=1}^{k-1} f_2(\overline{\sigma}\tilde{g}g_l)}{|\mathbb{S}_n|} = \sum_{g' \in \mathbb{S}_n} \frac{f_2(g') \bigotimes_{l=1}^{k-1} f_2(g'g_l)}{|\mathbb{S}_n|} = \mathcal{S}_{f_2}^{(k)}(G_{k-1})$. $\square$

**Intuition.** There are reasons to believe that computing higher correlations will increase the expressivity of the invariant. First, $k$-correlations are group-invariant polynomials of $f$ of order $k$. The Stone-Weierstrass theorem (Stone, 1948) tell us that polynomials are dense in continuous functions. Thus one can hope that the set of all $k$-correlations for arbitrary high $k$ will span a dense subset of the space of all invariants. In other words, one can hope that if a complete invariant exist, then it can be arbitrary well approximated by the $k$-correlations. Indeed, Adler & Konheim (1962) observed that correlation functions for $k = 1, 2, \dots$ form a complete translation invariant for real-valued, Haar-integrable functions on locally compact abelian groups. While (Chazan & Weiss, 1970) showed that in that general case the invariants are complete only for an infinite $k$, Kakarala (1992) references and shows settings for which 3-correlations are already complete. We conjecture that, for any finite $n$, there exists a finite $k_{\max}$ for which the collection of $k$-correlations, for all $k < k_{\max}$, forms a complete translation invariant for functions on $\mathbb{S}_n/\mathbb{S}_{n-2}$. In Section 4.1, we provide arguments to believe that $k_{\max} \in O(n^2)$.

Another confirmation arises from the study of higher correlations on undirected, unweighted graphs, in which case $A_{ij} \in \{0, 1\}$ and $A_{ii} = 0$. Here, higher correlation entries count increasingly larger graph substructures. For instance, $\mathcal{S}_f^{(1)} = \sum_{\tilde{g} \in \mathbb{S}_n} \frac{A_{\tilde{g}(n),\tilde{g}(n-1)}}{|\mathbb{S}_n|}$ counts the number of edges in the graph. Going on with similar calculations, $\mathcal{S}_f^{(2)}(g_1)$ recovers the results of $\mathcal{S}_f^{(1)}$ for $g_1 = ()$ and $g_2 = (n-1, n)$, counts the number of pairs of edges that share a common vertex for $g \in \{(n-2, n-1), (n-2, n), (n-2, n-1, n), (n-$

$2, n, n-1)\}$, and counts the number of pairs of edges that do not share a vertex for $g = (n-3, n-1)(n-2, n)$. Both the first and second correlations are sufficient to distinguish all graphs with up to three vertices. Finally, one can deduce that $\mathcal{S}_f^{(3)}((n-2, n-1), (n-3, n-1))$ is proportional to the number of triples of edges that all share one vertex, while $\mathcal{S}_f^{(3)}((n-2, n), (n-2, n-1))$ is proportional to number of triples of edges that form a triangle, allowing the invariant to distinguish graphs with more nodes.

### 4.1. Sparsity and double reduction

The increased expressivity of higher correlations comes with higher computational complexity. Indeed, (8) needs to be evaluated on $k-2$ group elements, each one taking up to $n!$ values. Here we prove a result that reduces the number of $k$-Spectrum entries to compute.

**Theorem 4.4.** *Let $f : \mathbb{S}_n / \mathbb{S}_{n-2} \to \mathbb{C}^d$. The Multi-Orbit $k$-Spectrum of $f$ is uniquely determined by its subset of components $\{\hat{\mathcal{S}}_f^{(k)}(G_{k-2}, \rho) | \rho \in \Lambda_n, G_{k-2} \in (\mathbb{S}_n / \mathbb{S}_{n-2})^{k-2}\}$.*

*Proof.* Observe Def. 4.2. To restrict $G_{k-2}$, it suffices to see that for any $H_{k-2} \in (\mathbb{S}_{n-2})^{k-2}$, $\prod_{l=1}^{k-2} f_{i_l}(\tilde{g} g_l h_l) = \prod_{l=1}^{k-2} f_{i_l}(\tilde{g} g_l)$. To restrict $\rho$, use Lemma 2.1. $\square$

While the Skew Spectrum and its Multi-Orbit generalization only need to be evaluated on 7 group elements ($\mathbb{S}_{n-2} \backslash \mathbb{S}_n / \mathbb{S}_{n-2}$) and 4 irreps ($\Lambda_n$), the $k$-Spectrum requires $O(n^{2(k-2)})$ group entries and 4 irreps.

To reduce the computational complexity, while maintaining the generalization, we employ two heuristics: the reduction of Kondor & Borgwardt (2008), and a double reduction that further limits the entries to evaluate.

**Reduction.** Analogously to the approach of Theorem 2.4, we define the reduced $k$-Spectrum as

$$\hat{\mathcal{S}}_f^{(k)}(G_{k-2}, \rho) = \bigoplus_{I_k} \hat{s}_{I_{k-1}}^{(k)}(G_{k-2}, \rho)^{\dagger} \hat{f}_{i_k}(\rho) \qquad (10)$$

where $\hat{s}_{I_{k-1}}^{(k)}(G_{k-2}, \rho) = \hat{r}_{I_{k-1}}^{(k)}(G_{k-2}, \rho) \tau_{\mathbb{S}_{n-2}}(\rho)$ and $\tau_{\mathbb{S}_{n-2}}(\rho)$ is the projection of $\rho$ onto the trivial irrep of $\mathbb{S}_{n-2}$.

$\tau_{\mathbb{S}_{n-2}}$ makes the rows of $\hat{s}_{I_{k-1}}^{(k)}(G_{k-2}, \rho)^{\dagger}$ match the sparsity patterns of the Fourier transform columns (Lemma 2.1 and Appendix B), making $\hat{s}^{(k)}$ easier to compute and restricting the non-zero elements of each the direct sum term to 49.

**Double reduction.** The Doubly-reduced $k$-Spectrum is given by the entries of the reduced $k$-Spectrum evaluated on $G_{k-2}$ taking values in $(\mathbb{S}_{n-2} \backslash \mathbb{S}_n / \mathbb{S}_{n-2})^{k-2}$, without repetitions or order. This heuristic reduces the number of possible entries from $n^{2(k-2)}$ to $\binom{7}{k-2}$. This effectively restricts the

possible values of $k$ to a maximum of 9, with the spectrum reaching its peak number of entries in the range $[4, 6]$. While this heuristics heavily sacrifices expressivity, we will show that it still adds completeness to the reduced Skew Spectrum, without increasing its computational complexity.

The reasons that motivate these heuristics are two. First, the Skew Spectrum and the Multi-Orbit version only need to be evaluated on the 7 double cosets representatives $\mathbb{S}_{n-2} \backslash \mathbb{S}_n / \mathbb{S}_{n-2}$. We extend this to $k$-Spectra by analogy.

Second, we note that repeated elements do not increase expressivity when representing unweighted graphs, and that the order of the elements in $G_{k-2}$ does not matter.

**Theorem 4.5** (Element distinctness). *Let $f : \mathbb{S}_n \to \{0, 1\}$ and let $G_{k-2}$ the list obtained by adding one element to $G_{k-3}$. Then, $\hat{\mathcal{S}}_f^{(k)}(G_{k-2}, \rho)$ adds information over $\hat{\mathcal{S}}_f^{(k-1)}(G_{k-3}, \rho)$ only if the added element in $G_{k-2}$ is not already in $G_{k-3}$.*

*Proof.* Because of the Fourier bijection, we consider $k$-correlations (see Def. 4.1). Define $\overline{G}_{k-1}$ by taking $G_{k-2}$ and adding one element $\sigma$, and then $\overline{G}_{k-2}$ by taking $G_{k-3}$ and adding the same $\sigma$. Using $f(g)^2 = f(g)$ and (7) one can verify $\mathcal{S}_f^{(k)}(\overline{G}_{k-1}) = \mathcal{S}_f^{(k-1)}(\overline{G}_{k-2})$, implying that $\hat{\mathcal{S}}_f^{(k)}(G_{k-2}, \rho) = \hat{\mathcal{S}}_f^{(k-1)}(G_{k-3}, \rho)$. $\square$

**Theorem 4.6** (Element ordering). *Let $f : \mathbb{S}_n \to \mathbb{C}$. $\hat{\mathcal{S}}_f^{(k)}(G_{k-2}, \rho)$ adds information over $\hat{\mathcal{S}}_f^{(k)}(G'_{k-2}, \rho)$ only if $G_{k-2}$ and $G'_{k-2}$ contain distinct sets of elements.*

*Proof.* Consider the $k$-Spectrum (Def. 4.2). Let $G'_{k-2}$ contain permuted elements of $G_{k-2}$. Since the terms $f(g g_l)$ commute, we have $\hat{r}^{(k)}(G_{k-2}, \rho) = \hat{r}^{(k)}(G'_{k-2}, \rho)$. $\square$

## 5. Relationship to WL hierarchy and GNNs

Understanding the expressive power of Graph Neural Networks has become a central theme in graph learning research. A key result in this area establishes that message-passing neural networks - a broad class of GNNs - are at most as expressive as the 1-dimensional Weisfeiler-Lehman (1-WL) graph isomorphism test (Xu et al., 2019; Morris et al., 2019). Recent work has extended this line of analysis to alternative architectures, including those based on relational pooling and spectral features, relating them to the broader Weisfeiler-Lehman hierarchy (Zhou et al., 2023; Zhang et al., 2024).

Within this landscape, exploring the relationship between our proposed $k$-Spectra (or $k$-correlations) framework and the WL hierarchy offers a promising avenue for both theoretical insight and practical application. Understanding this connection can shed light on the representational capabilities of $k$-Spectra and suggests ways it may enhance the

**Algorithm 1** Doubly-Reduced k-Spectrum

**Input:** Functions $f_i : \mathbb{S}_n/\mathbb{S}_{n-2} \to \mathbb{C}$, for $i \in [d]$
Precompute $s^{(k)}$
**for** $\rho \in \{(n), (n-1,1), (n-2,2), (n-2,1,1)\}$ **do**
  **for** $i \in [d]$ **do**
    Compute $\hat{f}_{i_k}(\rho)$
  **end for**
  **for** $G_{k-2} \in \mathrm{comb}(\mathbb{S}_{n-2}\backslash\mathbb{S}_n/\mathbb{S}_{n-2}, k-2)$ **do**
    **for** $I_{k-1} \in [d]^{k-1}$ **do**
      Compute $\hat{s}^{(k)}_{I_{k-1}}(G_{k-2}, \rho)$
    **end for**
    $\hat{\mathcal{S}}^{(k)}_f(G_{k-2}, \rho) = \bigoplus_{I_k \in [d]^k} \hat{s}^{(k)}_{I_{k-1}}(G_{k-2}, \rho)^\dagger \hat{f}_{i_k}(\rho)$
  **end for**
**end for**
Output $\hat{\mathcal{S}}^{(k)}_f$

expressivity of graph learning models, such as GNNs.

Theoretically, $k$-WL and $k$-correlation spectra capture distinct structural properties: while $k$-WL iteratively refines node representations based on patterns over $k$-tuples of node neighborhoods, $k$-correlations count edge-level substructures (see Intuition in the previous section). This distinction positions $k$-Spectra as a complementary framework for evaluating and augmenting expressivity.

As an illustrative example, Gai et al. (2025, Corollary 3.14) show that spectral-invariant GNNs can count paths and cycles of up to 7 vertices. In contrast, non-reduced $k$-Spectra - with sufficiently large $k$ - can capture longer paths and larger cycles, suggesting a path for practical advantage.

To empirically examine the relationship with WL tests, in Section 7.2 we demonstrate that features derived from the doubly-reduced $k$-Spectra are neither strictly less expressive nor strictly more expressive than those derived from 1-WL. This supports the intuition that $k$-Spectra offer orthogonal structural insights not captured by WL-based approaches.

Beyond theoretical comparisons with WL tests, we highlight three potential integration approaches into GNN pipelines:

- **Post concatenation:** Precompute a $k$-Spectra whole-graph embedding and concatenate it with the output of the GNN embedding, just before the prediction layers.
- **k-Spectra aggregation:** Incorporate $k$-Spectra sub-graph invariants during the node aggregation step.
- **k-Spectra pooling:** Replace standard pooling functions (e.g., summation, max) with $k$-Spectra-based invariants to aggregate node features.

Each of these strategies supports the incorporation of learnable parameters, such as orbit-specific weights (terms of Eq. 8), allowing the model to adaptively emphasize the most relevant graph information. We see these directions as

**Algorithm 2** Precomputing $s^{(k)}$

**Input:** Functions $f_i : \mathbb{S}_n/\mathbb{S}_{n-2} \to \mathbb{C}$, for $i \in [d]$
**for** $i \in [d]$ **do**
  **for** $j \in [n-2]$ **do**
    $F_i(j) = \sum_{l \in [n]\backslash\{j\}} f_i((n-1,j,n,l))$
    **for** $\sigma \in \mathbb{S}_n/\mathbb{S}_{n-2}$ **do**
      $P_i(g_1, \sigma, j) = f_i(\sigma)$
      $P_i(g_2, \sigma, j) = f_i(\sigma \cdot (n-1, n))$
      $P_i(g_3, \sigma, j) = f_i(\sigma \cdot (n-1, j))$
      $P_i(g_4, \sigma, j) = f_i(\sigma \cdot (n, j))$
      $P_i(g_5, \sigma, j) = f_i(\sigma \cdot (j, n-1, n))$
      $P_i(g_6, \sigma, j) = f_i(\sigma \cdot (j, n, n-1))$
      $P_i(g_7, \sigma, j) = \dfrac{F_i(\sigma(j)) - \sum_{l \in \{3,6\}} P_i(g_l, \sigma, j)}{n-3}$
    **end for**
  **end for**
**end for**
**for** $I_{k-2} \in [d]^{k-2}$ **do**
  **for** $G_{k-2} \in \mathrm{comb}(\mathbb{S}_{n-2}\backslash\mathbb{S}_n/\mathbb{S}_{n-2}, k-2)$ **do**
    **for** $\sigma \in \mathbb{S}_n/\mathbb{S}_{n-2}$ **do**
      $Q_{I_{k-2}}(G_{k-2}, \sigma) = \sum_{j=1}^{n-2} \prod_{l=1}^{k-2} P_{i_l}(g_l, \sigma, j)$
      **for** $i \in [d]$ **do**
        $I_{k-1} = \mathrm{append}(I_{k-2}, i)$
        $s^{(k)}_{I_{k-1}}(G_{k-2}, \sigma) = \frac{f_{i_{k-1}}(\sigma)}{n-2} Q_{I_{k-2}}(G_{k-2}, \sigma)$
      **end for**
    **end for**
  **end for**
**end for**
Output $s^{(k)}$

promising opportunities for future work at the intersection of spectral methods and GNN design.

## 6. Computation and complexity

We proceed to detail the computation of the Doubly-Reduced $k$-Spectrum (10), which is the generalization of the Skew Spectrum with the lowest computational complexity.

The main idea is summarized in Algorithm 1. We can treat $\hat{s}^{(k)}$ as the Fourier transform of a function $s^{(k)} \in \mathbb{S}_n/\mathbb{S}_{n-2}$.

$$s^{(k)}_{I_{k-1}}(G_{k-2}, \sigma) = \frac{f_{i_{k-1}}(\sigma)}{|\mathbb{S}_{n-2}|} \sum_{h \in \mathbb{S}_{n-2}} \prod_{l=1}^{k-2} f_{i_l}(\sigma h g_l) \quad (11)$$

After precomputing $s^{(k)}$, we can compute its transform and that of one single orbit function, for all the input. Then, multiplying them produces one entry of the Spectrum.

Precomputing $s^{(k)}$ efficiently requires some creativity. Algorithm 2 does so, using a divide-and-conquer strategy, coupled with dynamic programming. The procedure breaks the computation of $s^{(k)}$ into the computation of partial sums

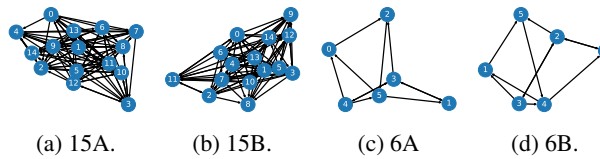

(a) 15A.    (b) 15B.    (c) 6A    (d) 6B.

*Figure 3.* Synthetic dataset with four graphs families. Each family contains 1000 isomorphic graphs. Families are not isomorphic.

that can be efficiently recombined when fixing the input parameters. We summarize this intermediate result in the following Lemma, discussing its correctness in Appendix C.

**Lemma 6.1** (Precomputing $s^{(k)}$). *Algorithm 2 precomputes $s^{(k)}_{I_{k-1}}(G_{k-2}, \sigma)$ for all the choices of the input parameters $I_{k-1} \in [d]^{k-1}$, $G_{k-2} \in (\mathbb{S}_{n-2}\backslash\mathbb{S}_n/\mathbb{S}_{n-2})^{k-2}$, $\sigma \in \mathbb{S}_n/\mathbb{S}_{n-2}$. It requires $O(d^{k-2}n^3 + d^{k-1}n^2)$ operations and $O(dn^3 + d^{k-1}n^2)$ space.*

Combining this lemma with the efficient computation of Fourier transforms on $\mathbb{S}_n/\mathbb{S}_{n-2}$, we arrive to the following.

**Theorem 6.2** (Doubly-reduced $k$-Spectrum). *Let $f$ : $\mathbb{S}_n/\mathbb{S}_{n-2} \to \mathbb{C}^d$. We can compute its doubly-reduced $k$-spectrum in $O(d^{k-1}n^3 + d^k n^2)$ operations. The doubly-reduced $k$-spectrum has up to $\binom{7}{k-2}7d^k$ non-zero entries.*

This result recovers the complexity of the reduced Multi-Orbit Skew Spectrum (Theorem 3.2) for $k = 3$ and that of the reduced Skew Spectrum (Theorem 2.4) for $d = 1$. The double reduction limits the scaling with $n$. On a single orbit, this enables the computation of more expressive invariants at the same computational cost of the Skew Spectrum.

### 6.1. Complexity analysis

We briefly prove Theorem 6.2, discussing the complexity of the algorithm and the number of non-zero entries.

**Main algorithm.** Algorithm 1 precomputes $s^{(k)}$ (Lemma 6.1) and then proceeds to compute the entries of $\hat{\mathcal{S}}^{(k)}_f$. The for loops compute $d^{k-1}$ Fourier transforms on $s^{(k)}$ and $d$ on $f_i$, each of which takes $O(n^3)$ operations, since both are functions on $\mathbb{S}_n/\mathbb{S}_{n-2}$. Finally, the direct sum performs $d^k$ matrix multiplication that cost $O(n^2)$ each (see the sparsity patterns of Lemma 2.1). The overall complexity is indeed $O(d^{k-1}n^3 + d^k n^2)$.

**Non-zero entries.** For each of the 4 irreps, there are $\binom{7}{k-2}$ lists $G_{k-2}$ on which evaluate the spectrum. Each spectrum entry has $d^k$ terms in the direct sum, with non-zero entries determined by the irrep. Using (10) and the column sparsity of the Fourier transform (Lemma 2.1), we observe that each $\rho \in \{(n), (n-1,1), (n-2,1,1)\}$ produces 1 non-zero entry, while $\rho = (n-2,2)$ produces 4: $(7 = 1 \cdot 3 + 4)$.

## 7. Numerical experiments

We implemented a proof-of-concept for the Doubly-reduced $k$-Spectrum to validate our theoretical insights. We report experiments that illustrate our contributions and show how the new results add expressivity to the original embedding.

### 7.1. Multiple orbits

First, we illustrate the potential of Multi-Orbit embeddings.

**Eigenvalue collision.** A first demonstration considers graphs with node labels, represented by adjacency matrices with labels encoded in the diagonal. While eigenvalues and singular values also form valid permutation-invariant embeddings (permutations act as similarity transformations through orthogonal matrices, preserving the spectrum), they fail to distinguish such structures.

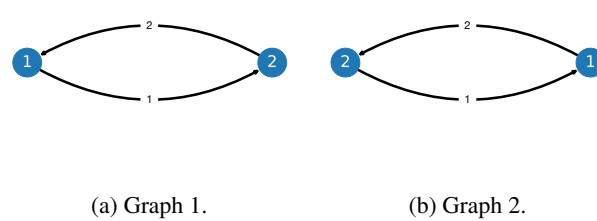

(a) Graph 1.      (b) Graph 2.

*Figure 4.* Two directed, weighted, labeled non-isomorphic graphs.

For example, the two graphs in Fig. 4 are non-isomorphic, as - for instance - the path from 1 to 2 has different weights. Their adjacency matrices, $A_1 = \begin{bmatrix} 1 & 1 \\ 2 & 2 \end{bmatrix}$ and $A_2 = \begin{bmatrix} 2 & 1 \\ 2 & 1 \end{bmatrix}$ share eigenvalues $\text{Eigs}(A_1) = \text{Eigs}(A_2) = \{3, 0\}$ and singular values $\{3.16, 0\}$. However, the reduced 2-Orbit Skew Spectrum, using $f_1(\sigma) = A_{\sigma(n),\sigma(n-1)}$ and $f_2(\sigma) = A_{\sigma(n),\sigma(n)}$, correctly distinguishes them.

**Synthetic graphs.** We evaluated the Multi-Orbit generalization on a synthetic dataset of labeled, directed, and weighted graphs. The dataset consists of four families: $(15A, 15B, 6A, 6B)$ (Fig. 3). Graphs within a family are isomorphic under edge- and label-preserving permutations, but graphs across different families are not. Specifically, in $15A$ and $15B$ (and similarly in $6A$ and $6B$), there is a permutation that links the edges and a separate permutation that links the labels, but these permutations are not the same. The task is to correctly classify the four families.

We processed the graphs using two representations: (1) the concatenation of Single-Orbit Skew Spectra computed on graph structure and labels, and (2) the 2-Orbit Skew Spectrum. A Random Forest classifier (60 estimators, no max depth) trained on 80% of the data achieves 50% accuracy

*Table 1.* Atomization energy regression on QM7, for different representations and models. Results in Mean Average Error (MAE).

| REPRESENTATION | XGB | GBR | EN | LINEAR |
|---|---|---|---|---|
| SINGLE-ORBIT | 29.15 | 36.55 | 114.68 | 61.15 |
| 2-ORBIT | **18.28** | 27.12 | 58.60 | 49.45 |
| C'S EIGS | 38.04 | 37.92 | 47.83 | - |
| LAPL.'S EIGS | 23.52 | 26.93 | 47.62 | 47.80 |

with the Single-Orbit representation and $100\%$ with the 2-Orbit representation. This is because the concatenation of Single-Orbit Skew Spectra cannot distinguish the couples $15A$-$15B$ and $6A$-$6B$, whose labels and edges are linked by different permutations, while the 2-Orbit Skew Spectra capture interactions between edges and labels (6).

**QM7.** We tested the Multi-Orbit generalization on the QM7 dataset, which contains Coulomb matrices of 7,165 molecules with up to 23 atoms (Rupp et al., 2012; Blum & Reymond, 2009). The Coulomb matrix $C \in \mathbb{R}^{23 \times 23}$ is defined as $C_{ii} = \frac{1}{2}Z_i^{2.4}$ and $C_{ij} = \frac{Z_i Z_j}{|R_i - R_j|}$, where $Z_i$ is the nuclear charge of the $i$-th atom of the molecule, and $R_i$ is its position. The task is to predict molecular atomization energies (kcal/mol), renormalized from $-1$ to $1$.

We compare different representations: the Single-Orbit Skew Spectrum on the off-diagonals, the 2-Orbit Skew Spectrum, and eigenvalues of adjacency and Laplacian matrices. We train several models on a $80\%$-$20\%$ split: Extreme Gradient Boosting (XGB), Gradient Boosting Regressor (GBR), Elastic Net (EN), Linear Regression (Linear) (Pedregosa et al., 2011). Table 1 reports the results, with XGB and the 2-Orbit Skew Spectrum achieving the best performance.

### 7.2. Higher correlations

We study how higher-order correlations affect expressivity using two datasets of non-isomorphic, unweighted, undirected graphs: the Atlas of all graphs with 7 nodes, and a set of connected chordal graphs with 8 nodes (McKay). For each dataset, we computed the Doubly-Reduced $k$-Spectra for $k \in [3, 9]$, their concatenation, the eigenvalues of the graph Laplacian, and the 1-WL test for multiple iterations. We then measured the number of collisions, i.e., cases where non-isomorphic graphs could not be distinguished by each method. Figure 5 summarizes the results.

Increasing $k$ significantly improves the expressivity of the Skew Spectrum ($k = 3$), reducing the number of collisions. For $n = 7$, the concatenated $k$-Spectra form a complete invariant - no collisions remain - outperforming both the Laplacian eigenvalues and 1-WL in distinguishing graph structures. Interestingly, 1-WL does not achieve a complete invariant for the $n = 7$ graphs, and the concatenated

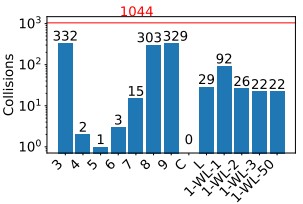

(a) Graph Atlas, 7 nodes.

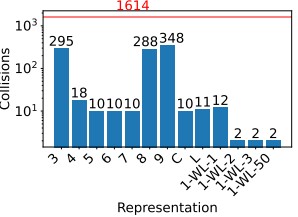

(b) Chordal graphs, 8 nodes.

*Figure 5.* Shattering non-isomorphic graphs with $k$-Spectra ($k \in [3, 9]$), their concatenation (C), Laplacian's eigenvalues (L), and 1-WL tests with different iterations (1-WL-ITER).

$k$-Spectra offer greater expressivity in that setting. On the other hand, for the chordal graphs with $n = 8$, 1-WL results in fewer collisions than the concatenated spectra. This outcome supports the perspective discussed in Section 5, reinforcing the idea that $k$-correlations and WL tests measure different structural properties and can serve as complementary expressivity metrics.

## 8. Conclusion

This work generalizes the Skew Spectrum (Kondor & Borgwardt, 2008), advancing group-theoretic methods for permutation-invariant graph embeddings. We introduced *Multi-Orbit Spectra* for complex graph structures, *$k$-Spectra* to trade-off expressivity and complexity, and *Doubly-Reduced $k$-Spectra* to enhance expressivity of the original Skew Spectrum without added cost. We support our theoretical insights with numerical experiments. This study complements modern approaches and opens new avenues for research in graph representation.

**Future directions.** On the *theoretical side*, (1) a key open question is whether there exists a finite $k_{max}$ such that the collection of all $k$-correlations up to $k_{max}$ forms a complete invariant, which remains to be proven. (2) Our framework, focused on $\mathbb{S}_n / \mathbb{S}_{n-2}$, could naturally be extended to functions on $G/H$ for arbitrary groups/subgroups, broadening its scope beyond graphs to invariants for other structured data domains. (3) Another avenue is exploring other families of translation-invariant polynomials beyond $k$-correlations.

On the *practical side*, (1) developing a scalable, optimized open-source implementation with thorough benchmarking is a crucial step toward real-world adoption. (2) Integrating our methods with modern deep learning frameworks offers further opportunities: (2.1) introducing trainable parameters, such as learning the relative importance of correlations between orbits, or (2.2) using the generalized Skew Spectrum as a positional encoding, aggregation, or pooling function to enhance neural representations.

## Impact Statement

This paper presents work whose goal is to advance the field of Machine Learning. There are many potential societal consequences of our work, none which we feel must be specifically highlighted here.

## Acknowledgements

First and foremost, we would like to thank Ramakrishna Kakarala for providing us with a soft copy of his thesis. We also extend our sincere thanks to the Quantum Open Source Foundation (QOSF), through which the authors met and this project started.

A.B. gratefully acknowledges Stefano Gogioso and Erickson Tjoa for insightful discussions that helped improve the mathematical notation; Filippo Guerranti for providing references, background material, and assistance in revising the manuscript and supporting the submission process; and Alessandro Palma for valuable feedback on experiments and the submission process. He also thanks Professors Stefano Zanero, Donatella Sciuto, and Ferruccio Resta for their support during his Ph.D., Patrick Rebentrost for hosting him at CQT during part of this research, and Professor Ignacio Cirac for his support during his postdoctoral work. A.B.'s research was partially funded by THEQUCO as part of the Munich Quantum Valley, supported by the Bavarian State Government through the Hightech Agenda Bayern Plus. Additional financial support was provided by ICSC - "National Research Centre in High Performance Computing, Big Data and Quantum Computing," Spoke 10, funded by the European Union - NextGenerationEU, under grant *PNRR-CN00000013-HPC*.

M.P. acknowledges support from the Deutsche Forschungsgemeinschaft (DFG, German Research Foundation, project numbers 447948357 and 440958198), the Sino-German Center for Research Promotion (Project M-0294), the German Ministry of Education and Research (Project QuKuK, BMBF Grant No. 16KIS1618K), the DAAD, the Alexander von Humboldt Foundation, and the Niedersächsisches Ministerium für Wissenschaft und Kultur.

A.L. acknowledges the support of the National Research Foundation, Singapore, and A*STAR under its Centre for Quantum Technologies Funding Initiative (S24Q2d0009), and the Quantum Engineering Programme (QEP 2.0) under grants *NRF2021-QEP2-02-P05* and *NRF2021-QEP2-02-P01*.

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

# A. The Symmetric Group $\mathbb{S}_n$

We review key concepts related to the symmetric group $\mathbb{S}_n$ to support our discussion. For a more in-depth treatment, see Grillet (2007), Rotman (1999), Raczka & Barut (1986), and the background sections of Bellante (2024).

The Symmetric Group $\mathbb{S}_n$ is the group of all permutations of $n$ objects. Each permutation $g \in \mathbb{S}_n$ acts by mapping the indices of objects in an ordered set to new positions, effectively defining a bijective function $g : [n] \to [n]$. The group has $|\mathbb{S}_n| = n!$ elements, corresponding to all possible rearrangements of $n$ indices.

Besides the cycle notation used in the main text, permutations can be conveniently represented as arrays:

$$
g = \begin{bmatrix} 1 & 2 & \cdots & n-1 & n \\ g(1), & g(2), & \cdots & g(n-1), & g(n) \end{bmatrix} \tag{12}
$$

where $g(i)$ denotes the image of index $i$ under $g$. For instance, the transposition swapping the first two elements in $\mathbb{S}_4$ is
$g = \begin{bmatrix} 1 & 2 & 3 & 4 \\ 2, & 1, & 3, & 4 \end{bmatrix}$.

This notation makes it easy to identify a left coset transversal for $\mathbb{S}_n/\mathbb{S}_{n-2}$ as a set containing one representative for each group element that fixes the last two positions $g = \begin{bmatrix} 1 & \cdots & n-1 & n \\ \cdots, & \cdots, & i, & j \end{bmatrix}$. Since $i, j \in [n]$ and $i \neq j$, there are $n(n-1)$ such representatives, giving $|\mathbb{S}_n/\mathbb{S}_{n-2}| = n(n-1)$.

Functions on graphs, such as $f(\sigma) = A_{\sigma(n),\sigma(n-1)}$, depend only on the image of the last two indices. This makes it clear that

$$
f(\sigma h) = f(\sigma), \text{for all } \sigma \in \mathbb{S}_n, h \in \mathbb{S}_{n-2}, \tag{13}
$$

since elements $h \in \mathbb{S}_{n-2}$ preserve the last two indices: $h = \begin{bmatrix} 1 & \cdots & n-1 & n \\ \cdots, & \cdots, & n-1, & n \end{bmatrix}$.

## A.1. Irreducible representations

We introduce the concepts that are most relevant to our work. For a more comprehensive treatment of the irreducible representations of $\mathbb{S}_n$, see Sagan (2013), Zhao (2008), and the background sections of Bellante (2024).

For our focus, a representation of $\mathbb{S}_n$ is a mapping $\rho : \mathbb{S}_n \to U(\dim(\rho))$ that assigns each permutation $\sigma \in \mathbb{S}_n$ a unitary matrix of size $\dim(\rho) \times \dim(\rho)$. This mapping satisfies the following properties:

1. $\rho(e) = I$, where $e$ is the identity permutation and $I$ the identity matrix.
2. $\rho(\sigma_1 \sigma_2) = \rho(\sigma_1)\rho(\sigma_2)$ for all $\sigma_1, \sigma_2 \in \mathbb{S}_n$.
3. $\rho(\sigma^{-1}) = \rho(\sigma)^\dagger$.

Since the direct sum of two representations is also a representation, we say that a representation is irreducible (irrep) if it cannot be decomposed into a direct sum of smaller representations. That is, if $\rho(\sigma) = \rho_1(\sigma) \oplus \rho_2(\sigma)$ for all $\sigma \in \mathbb{S}_n$, then either $\dim(\rho_1) = 0$ or $\dim(\rho_2) = 0$.

Irreducible representations of $\mathbb{S}_n$ are classified by partitions of $n$, up to equivalence under invertible linear transformations. A partition of $n$ is a sequence $\lambda = (\lambda_1, \lambda_2, \ldots, \lambda_l)$ of weakly decreasing positive integers ($\lambda_i \geq \lambda_{i+1}$) that sum to $n$. We denote this by $\lambda \vdash n$ and often use $\rho$ interchangeably for both the partition and the corresponding representation. The trivial partition of $n$ is $(n)$, which corresponds to the trivial representation $\rho : \mathbb{S}_n \to \mathbb{C}$ given by $\rho(\sigma) = 1$ for all $\sigma$.

A convenient way to visualize partitions is through Ferres diagrams (or Young diagrams), which consist of rows of boxes, where the $i$-th row contains $\lambda_i$ boxes. Because $\lambda$ is weakly decreasing, each row contains at most as many boxes as the row above. For example, the partition $(7, 4, 3, 1) \vdash 15$ is represented as:

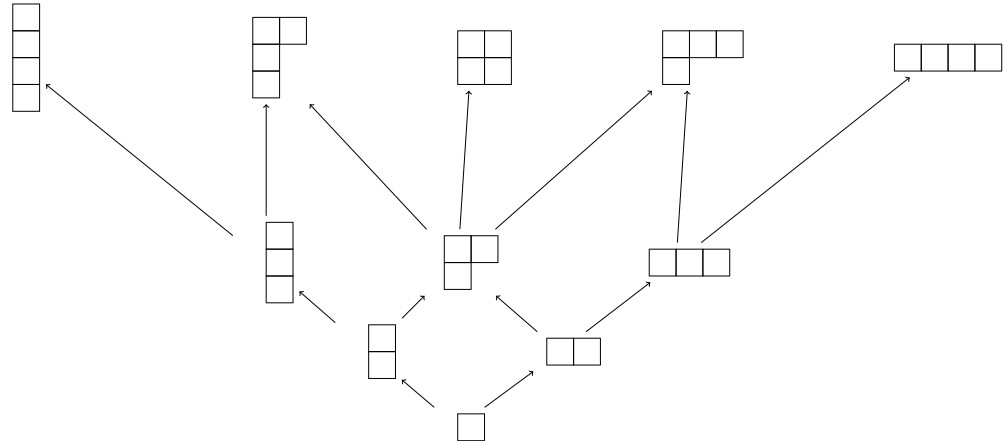

*Figure 6.* Young lattice for $\mathbb{S}_4$.

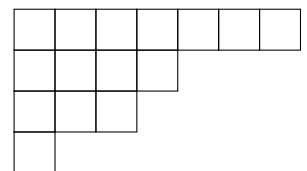

A Young tableau is a Young diagram filled with the numbers 1 through $n$ without repetition. A standard Young tableau has strictly increasing numbers along each row and each column. The dimension of an irrep labeled by $\lambda$ is given by the number of standard Young tableaux of shape $\lambda$. This number can be computed through the hook lenght formula. Table 2 lists the dimensions of irreps relevant to our work.

*Table 2.* Dimensions of irreducible representations of $\mathbb{S}_n$ indexed by partitions $\lambda \vdash n$. The dimension is given by the hook-length formula $f^\lambda$, which counts the number of standard tableaux of shape $\lambda$. These values are from Kondor (2008, Table 1.1).

| $\lambda$ | $f^\lambda$ or $\dim(\rho_\lambda)$ |
|:---:|:---:|
| $(n)$ | 1 |
| $(n-1,1)$ | $n-1$ |
| $(n-2,2)$ | $\frac{n(n-3)}{2}$ |
| $(n-2,1,1)$ | $\frac{(n-1)(n-2)}{2}$ |

In this work, we use the Young Orthogonal Representation (YOR), a specific irrep of $\mathbb{S}_n$ with real-valued, sparse matrices. This representation is adapted to the subgroup chain $\mathbb{S}_n \to \mathbb{S}_{n-1} \to \cdots \to \mathbb{S}_1$, meaning that any $\rho \vdash k$, when restricted to $\mathbb{S}_{k-1}$, decomposes as:

$$\rho \downarrow_{\mathbb{S}_{k-1}} (h) = \bigoplus_{\eta : \eta < \rho} \eta(h), \quad h \in \mathbb{S}_{k-1}. \tag{14}$$

Here $\rho \vdash k$, $\eta \vdash k-1$, and the notation $\eta < \rho$ means that $\eta$ can be obtained by removing one box from the Young diagram of $\rho$. This structure provides an efficient way to decompose irreps of $\mathbb{S}_n$ when acting on elements of a subgroup $\mathbb{S}_{n-k}$. Figure 6 illustrates this decomposition for $\mathbb{S}_4$. Such diagrams are often termed Bratteli diagrams or Young diagrams.

Finally, a useful result in representation theory, sometimes referred to as the Great Orthogonality Theorem (Zee (2016, Section II.3)), states that for any two irreps $\rho_1, \rho_2$,

$$\sum_{g \in \mathbb{S}_n} \rho_1(g)_{i,j}^\dagger \rho(g)_{k,l} = \frac{|\mathbb{S}_n|}{\dim(\rho)} \delta_{\rho_1,\rho_2} \delta_{i,l} \delta_{j,k}. \tag{15}$$

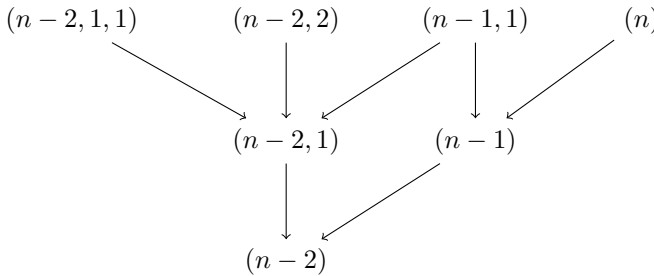

*Figure 7.* Partial Young lattice of the partitions of $n$ reaching $(n-2)$.

Here, $\delta_{\rho_1,\rho_2}$ is the Kronecker delta, which is 1 if $\rho_1$ and $\rho_2$ are equivalent (i.e., represent the same irrep) and 0 otherwise. A direct consequence of (15) is that summing over all elements of $\mathbb{S}_n$ in a given irrep of $\rho \vdash n$ yields:

$$\sum_{\sigma \in \mathbb{S}_n} \rho(\sigma) = \begin{cases} |\mathbb{S}_n| & \rho = (n) \\ 0 & \text{otherwise.} \end{cases} \tag{16}$$

This follows from the fact that the trivial representation $(n)$ has dimension 1 and consists entirely of ones.

## B. Fourier sparsity and projecting on the trivial irrep

In this appendix, we examine the role of $\tau_{\mathbb{S}_{n-2}}(\rho)$, which represents the projection of an irrep $\rho \in \mathrm{Irr}(\mathbb{S}_n)$ onto the trivial representation of $\mathbb{S}_{n-2}$:

$$\tau_{\mathbb{S}_{n-2}}(\rho) = \frac{1}{|\mathbb{S}_{n-2}|} \sum_{h \in \mathbb{S}_{n-2}} \rho(h) \tag{17}$$

This projection naturally arises in the Fourier transform of functions $\mathbb{S}_n/\mathbb{S}_{n-2} \to \mathbb{C}$ (Kondor, 2007):

$$\hat{f}(\rho) = \frac{1}{|\mathbb{S}_n|} \sum_{\sigma \in \mathbb{S}_n} f(\sigma)\rho(\sigma) = \frac{1}{|\mathbb{S}_n|} \sum_{y \in \mathbb{S}_n/\mathbb{S}_{n-2}} \sum_{h \in \mathbb{S}_{n-2}} f(yh)\rho(yh) \tag{18}$$

$$= \frac{|\mathbb{S}_{n-2}|}{|\mathbb{S}_n|} \sum_{y \in \mathbb{S}_n/\mathbb{S}_{n-2}} f(y)\rho(y) \frac{1}{|\mathbb{S}_{n-2}|} \sum_{h \in \mathbb{S}_{n-2}} \rho(h) \tag{19}$$

$$= \frac{|\mathbb{S}_{n-2}|}{|\mathbb{S}_n|} \sum_{y \in \mathbb{S}_n/\mathbb{S}_{n-2}} f(y)\rho(y)\tau_{\mathbb{S}_{n-2}}(\rho). \tag{20}$$

Moreover, $\tau_{\mathbb{S}_{n-2}}(\rho)$ is responsible for the sparsity patterns described in Lemma 2.1.

Using the fact that YOR is adapted (14) , along with a special case of the Great Orthogonality Theorem (16), we obtain:

$$\tau_{\mathbb{S}_{n-2}}(\rho) = \frac{1}{|\mathbb{S}_{n-2}|} \bigoplus_{\eta < \rho} \sum_{h \in \mathbb{S}_{n-2}} \eta(h) = \bigoplus_{\eta < \rho} \begin{cases} 1 & \text{if } \eta = (n-2) \\ 0 & \text{otherwise.} \end{cases} \tag{21}$$

This shows that $\tau_{\mathbb{S}_{n-2}}(\rho)$ is a diagonal matrix containing mostly 0s, except for a few 1s corresponding to the positions where $(n-2)$ appears in the decomposition of $\rho$.

From the structure of the Young lattice, we see that there are exactly four partitions of $n$ that contain $(n-2)$:

$$(n), (n-1,1), (n-2,2), (n-2,1,1). \tag{22}$$

This set of partitions is refereed to as $\Lambda_n$ in the main text. Figure 7 provides a visualization of a partial Young lattice, illustrating these decompositions.

Since all other irreps do not contain $(n-2)$, their projection $\tau_{\mathbb{S}_{n-2}}(\rho)$ is the zero matrix, implying that their Fourier transform also vanishes.

Further observation reveals that three of these partitions - $(n), (n-1, 1), (n-2, 1, 1)$ - contain $(n-2)$ exactly once, while $(n-2, 2)$ contains it twice. The number of paths from $\rho$ to $(n-2)$ in the Young lattice determines how many diagonal entries of $\tau_{\mathbb{S}_{n-2}}(\rho)$ are 1.

Combining this observation with the dimensions of the irreps (see Table 2) completes the proof of Lemma 2.1.

Finally, note that, since $\tau_{\mathbb{S}_{n-2}}(\rho)$ is a diagonal matrix, it follows that $\tau_{\mathbb{S}_{n-2}}(\rho) = \tau_{\mathbb{S}_{n-2}}(\rho)^{\dagger}$.

## C. Precomputing $s^{(k)}$ with Dynamic Programming

In this Appendix, we prove the correctness of Lemma 6.1 and Algorithm 2.

We begin by proving the following theorem.

**Theorem C.1.** *The Fourier transform of*

$$s_{I_{k-1}}^{(k)}(G_{k-2}, \sigma) = \frac{f_{i_{k-1}}(\sigma)}{|\mathbb{S}_{n-2}|} \sum_{h \in \mathbb{S}_{n-2}} \prod_{l=1}^{k-2} f_{i_l}(\sigma h g_l) \tag{23}$$

*over the last entry produces* $\hat{s}_{I_{k-1}}^{(k)}(G_{k-2}, \rho) = \hat{r}_{I_{k-1}}^{(k)}(G_{k-2}, \rho)\tau_{\mathbb{S}_{n-2}}(\rho)$, *where* $\hat{r}_{I_{k-1}}^{(k)}(G_{k-2}, \rho) = \frac{1}{|\mathbb{S}_n|} \sum_{\tilde{g} \in \mathbb{S}_n} f_{i_{k-1}}(\tilde{g}) \prod_{l=1}^{k-2} f_{i_l}(\tilde{g}g_l)\rho(\tilde{g})$ *and* $\tau_{\mathbb{S}_{n-2}}(\rho)$ *is the projection of* $\rho$ *onto the trivial irrep* $(n-2)$.

*Proof.* We compute $\hat{s}^{(k)}$ using the expressions from (23) and (3):

$$\hat{s}_{I_{k-1}}^{(k)}(G_{k-2}, \rho) = \frac{1}{|\mathbb{S}_n|} \sum_{\bar{g} \in \mathbb{S}_n} \frac{f_{i_{k-1}}(\bar{g})}{|\mathbb{S}_{n-2}|} \sum_{h \in \mathbb{S}_{n-2}} \prod_{l=1}^{k-2} f_{i_l}(\bar{g}hg_l)\rho(\bar{g}) \qquad (\text{use } \tilde{g} = \bar{g}h) \quad (24)$$

$$= \frac{1}{|\mathbb{S}_n|} \frac{1}{|\mathbb{S}_{n-2}|} \sum_{\tilde{g} \in \mathbb{S}_n} \sum_{h \in \mathbb{S}_{n-2}} f_{i_{k-1}}(\tilde{g}h^{-1}) \prod_{l=1}^{k-2} f_{i_l}(\tilde{g}g_l)\rho(\tilde{g}h^{-1}) \tag{25}$$

$$= \frac{1}{|\mathbb{S}_n|} \sum_{\tilde{g} \in \mathbb{S}_n} f_{i_{k-1}}(\tilde{g}) \prod_{l=1}^{k-2} f_{i_l}(\tilde{g}g_l)\rho(\tilde{g}) \left( \frac{1}{|\mathbb{S}_{n-2}|} \sum_{h \in \mathbb{S}_{n-2}} \rho(h)^{\dagger} \right) \qquad (\text{use } \tau_{\mathbb{S}_{n-2}}(\rho) = \tau_{\mathbb{S}_{n-2}}(\rho)^{\dagger}) \quad (26)$$

$$= \hat{r}_{I_{k-1}}^{(k)}(G_{k-2}, \rho)\tau_{\mathbb{S}_{n-2}}(\rho). \tag{27}$$

$\square$

It is also important to observe that $s_{I_{k-1}}^{(k)}(G_{k-2}, \sigma) = s_{I_{k-1}}^{(k)}(G_{k-2}, \sigma h)$ for any $h \in \mathbb{S}_{n-2}$. This implies that one can use a fast Fourier transform on $\mathbb{S}_n/\mathbb{S}_{n-2}$ to obtain $\hat{s}_{I_{k-1}}^{(k)}(G_{k-2}, \rho)$.

Now, we address the task of efficiently precomputing (23) for $i_1, \ldots, i_{k-1} \in [d], g_1, \ldots, g_{k-2}$ taking values in the double coset transversal $\mathbb{S}_{n-2} \backslash \mathbb{S}_n / \mathbb{S}_{n-2} = \{(), (n-1, n), (n-2, n-1), (n-2, n), (n-2, n-1, n), (n-2, n, n-1), (n-3, n-1)(n-2, n)\}$, and $\sigma \in \mathbb{S}_n/\mathbb{S}_{n-2}$.

Naively computing (23) for all the possible entries would be prohibitively expensive. Specifically, the sum over $\mathbb{S}_{n-2}$ contains $(n-2)!$ terms, making this computation infeasible even for small $n$. To circumvent this, we employ a divide-and-conquer strategy plus dynamic programming.

The main idea is to decompose (23) as

$$s_{I_{k-1}}^{(k)}(G_{k-2}, \sigma) = \frac{f_{i_{k-1}}(\sigma)}{n-2} Q_{I_{k-2}}(G_{k-2}, \sigma) \tag{28}$$

where $Q_{I_{k-2}}(G_{k-2}, \sigma) = \sum_{j=1}^{n-2} \prod_{l=1}^{k-2} P_{i_l}(g_l, \sigma, j)$ and we have 7 functions $P$, one per element in the double coset

1. $P_i(g_1, \sigma, j) = f_i(\sigma)$.
2. $P_i(g_2, \sigma, j) = f_i(\sigma \cdot (n-1, n))$.
3. $P_i(g_3, \sigma, j) = f_i(\sigma \cdot (n-1, j))$.
4. $P_i(g_4, \sigma, j) = f_i(\sigma \cdot (n, j))$.
5. $P_i(g_5, \sigma, j) = f_i(\sigma \cdot (j, n-1, n))$.
6. $P_i(g_6, \sigma, j) = f_i(\sigma \cdot (j, n, n-1))$.
7. $P_i(g_7, \sigma, j) = \dfrac{F_i(\sigma(j)) - \sum_{l \in \{3,6\}} P_i(g_l, \sigma, j)}{n-3}$

with $F_i(j) = \sum_{\substack{l \in [n] \\ l \neq j}} f_i((n-1, j, n, l))$.

Here, $\sigma \cdot g$ represents the group operation combining two elements, while $\sigma(j)$ denotes the image of $j$ under $\sigma$.

By utilizing dynamic programming, we can compute these $P-$functions efficiently and precompute $s^{(k)}$ without resorting to a naive approach.

We break Lemma 6.1 into two propositions, proving its complexity and correctness separately.

## C.1. Complexity

**Proposition C.2.** *Algorithm 2 requires $O(d^{k-2}n^3 + d^{k-1}n^2)$ operations and $O(dn^3 + d^{k-1}n^2)$ space.*

*Proof.* We analyze the complexity of each step in Algorithm 2.

- Computing $F_i(j)$: Each function $F_i(j)$ requires $O(n)$ operations. Since we need to compute $O(dn)$ such functions, the total time and space complexity for this step is $O(dn^2)$.
- Computing $P_i(g, \sigma, j)$: Each function $P_i(g, \sigma, j)$ can be evaluated in constant time, $O(1)$. We need to evaluate $O(dn^3)$ such functions, leading to a total time and space complexity of $O(dn^3)$.
- Computing $Q_{I_{k-2}}(G_{k-2}, \sigma)$: For a fixed constant $k \in [3, 9]$, each $Q_{I_{k-2}}(G_{k-2}, \sigma)$ can be evaluated in $O(n)$ operations. There are $d^{k-2}\binom{7}{k-2}n(n-1)$ entries, so the total time complexity of computing all the functions is $O(d^{k-2}n^3)$, and the space complexity is $O(d^{k-2}n^2)$.
- Computing $s^{(k)}_{I_{k-1}}(G_{k-2}, \sigma)$: Each $s^{(k)}_{I_{k-1}}(G_{k-2}, \sigma)$ can be evaluated in constant time, $O(1)$. There are $O(d^{k-1}n^2)$ such entries, requiring $O(d^{k-1}n^2)$ time and space.

Summing up the asymptotic complexities for time and space we obtain $O(d^{k-2}n^3 + d^{k-1}n^2)$ operations and $O(dn^3 + d^{k-1}n^2)$ space. $\qquad\square$

## C.2. Correctness

**Proposition C.3.** *Algorithm 2 precomputes $s^{(k)}_{I_{k-1}}(G_{k-2}, \sigma)$ for all the choices of the input parameters $I_{k-1} \in [d]^{k-1}$, $G_{k-2} \in (\mathbb{S}_{n-2} \backslash \mathbb{S}_n / \mathbb{S}_{n-2})^{k-2}$, $\sigma \in \mathbb{S}_n / \mathbb{S}_{n-2}$.*

*Proof.* The core idea behind the correctness of Algorithm 2 lies in breaking down the sum over $\mathbb{S}_{n-2}$. In particular, the crucial fact is that the sum over $\mathbb{S}_{n-2}$

$$\frac{1}{|\mathbb{S}_{n-2}|} \sum_{h \in \mathbb{S}_{n-2}} \prod_{l=1}^{k-2} f_{i_l}(\sigma h g_l) \tag{29}$$

can be broken as a sum of $n-2$ partial sums

$$\frac{1}{n-2} \sum_{j=1}^{n-2} \prod_{l=1}^{k-2} P_{i_l}(g_l, \sigma, j). \tag{30}$$

To illustrate this, we study the term

$$\frac{1}{|\mathbb{S}_{n-2}|} \sum_{h \in \mathbb{S}_{n-2}} f_i(\sigma h g) \tag{31}$$

for all $g \in \mathbb{S}_{n-2}\backslash\mathbb{S}_n/\mathbb{S}_{n-2}$ and show how it naturally breaks into $n-2$ partial sums indexed by $j \in [n-2]$. Specifically, we show that each $j$ corresponds to the $(n-3)!$ elements $h \in \mathbb{S}_{n-2}$ such that $h(n-2) = j$. With this additional fact, analyzing (31) suffices to prove the equivalence of (29) and (31).

In the following, for each of the 7 group elements, we consider the action of $\sigma h g$ on $n$ and $n-1$ and simplify the expression. For an ease of notation, we rewrite each $f_i$ as $f_i(\sigma) = f_i'(\sigma(n-1), \sigma(n))$, with $f_i' : [n]^2 \to \mathbb{C}$.

*1. $g_1 = ()$*

$$\sigma h g(n) = \sigma h(n) = \sigma(n) \tag{32}$$
$$\sigma h g(n-1) = \sigma h(n-1) = \sigma(n-1). \tag{33}$$

Hence,

$$\frac{1}{|\mathbb{S}_{n-2}|} \sum_{h \in \mathbb{S}_{n-2}} f_i'(\sigma h g(n-1), \sigma h g(n)) = f_i'(\sigma(n-1), \sigma(n)) = f_i(\sigma) \tag{34}$$

So the term breaks in $\frac{1}{n-2}\sum_{j=1}^{n-2} P_i(g_1, \sigma, j)$. The choice of how $j$ divides $\mathbb{S}_{n-2}$ is arbitrary here, so we can satisfy $h(n-2) = j$.

*2. $g_2 = (n-1, n)$*

$$\sigma h g(n) = \sigma h(n-1) = \sigma(n-1) \tag{35}$$
$$\sigma h g(n-1) = \sigma h(n) = \sigma(n) \tag{36}$$

Therefore,

$$\frac{1}{|\mathbb{S}_{n-2}|} \sum_{h \in \mathbb{S}_{n-2}} f_i'(\sigma h g(n-1), \sigma h g(n)) = f'(\sigma(n), \sigma(n-1)) = f(\sigma \cdot (n-1, n)). \tag{37}$$

So the term breaks in $\frac{1}{n-2}\sum_{j=1}^{n-2} P_i(g_2, \sigma, j)$. Similarly to the previous case, the choice of how $j$ divides $\mathbb{S}_{n-2}$ is arbitrary, so we satisfy $h(n-2) = j$.

*3. $g_3 = (n-2, n-1)$*

$$\sigma h g(n) = \sigma h(n) = \sigma(n) \tag{38}$$
$$\sigma h g(n-1) = \sigma h(n-2) = \sigma(j) \text{ for } j \in [n-2] \tag{39}$$

There are $(n-3)!$ elements $h \in \mathbb{S}_{n-2}$ that can produce a certain $j$, and there are $n-2$ possible $j$. Grouping these terms, we can write

$$\frac{1}{|\mathbb{S}_{n-2}|} \sum_{h \in \mathbb{S}_{n-2}} f_i'(\sigma h g(n-1), \sigma h g(n)) = \frac{1}{(n-2)!} \sum_{j=1}^{n-2}(n-3)! f'(\sigma(j), \sigma(n)) = \frac{1}{n-2}\sum_{j=1}^{n-2} f(\sigma \cdot (n-1, j)). \tag{40}$$

So the term breaks in $\frac{1}{n-2}\sum_{j=1}^{n-2} P_i(g_3, \sigma, j)$.

*4. $g_4 = (n-2, n)$*

$$\sigma h g(n) = \sigma h(n-2) = \sigma(j) \text{ for } j \in [n-2] \tag{41}$$
$$\sigma h g(n-1) = \sigma h(n-1) = \sigma(n-1) \tag{42}$$

Similarly to the case above,

$$\frac{1}{|\mathbb{S}_{n-2}|} \sum_{h \in \mathbb{S}_{n-2}} f_i'(\sigma h g(n-1), \sigma h g(n)) = \frac{1}{(n-2)!} \sum_{j=1}^{n-2}(n-3)! f'(\sigma(n-1), \sigma(j)) = \frac{1}{n-2}\sum_{j=1}^{n-2} f(\sigma \cdot (n, j)). \tag{43}$$

So the term breaks in $\frac{1}{n-2}\sum_{j=1}^{n-2} P_i(g_4, \sigma, j)$.

5. $g_5 = (n-2, n-1, n)$

$$\sigma h g(n) = \sigma h(n-2) = \sigma(j) \text{ for } j \in [n-2] \tag{44}$$
$$\sigma h g(n-1) = \sigma h(n) = \sigma(n) \tag{45}$$

Hence,

$$\frac{1}{|\mathbb{S}_{n-2}|} \sum_{h \in \mathbb{S}_{n-2}} f_i'(\sigma h g(n-1), \sigma h g(n)) = \frac{1}{(n-2)!} \sum_{j=1}^{n-2} (n-3)! f'(\sigma(n), \sigma(j)) = \frac{1}{n-2} \sum_{j=1}^{n-2} f(\sigma \cdot (j, n-1, n)). \tag{46}$$

So the term breaks in $\frac{1}{n-2}\sum_{j=1}^{n-2} P_i(g_5, \sigma, j)$.

6. $g_6 = (n-2, n, n-1)$

$$\sigma h g(n) = \sigma h(n-1) = \sigma(n-1) \tag{47}$$
$$\sigma h g(n-1) = \sigma h(n-2) = \sigma(j) \text{ for } j \in [n-2] \tag{48}$$

Therefore,

$$\frac{1}{|\mathbb{S}_{n-2}|} \sum_{h \in \mathbb{S}_{n-2}} f_i'(\sigma h g(n-1), \sigma h g(n)) = \frac{1}{(n-2)!} \sum_{j=1}^{n-2} (n-3)! f'(\sigma(j), \sigma(n-1)) \tag{49}$$

$$= \frac{1}{n-2} \sum_{j=1}^{n-2} f(\sigma \cdot (j, n, n-1)). \tag{50}$$

So the term breaks in $\frac{1}{n-2}\sum_{j=1}^{n-2} P_i(g_6, \sigma, j)$.

7. $g = (n-3, n-1)(n-2, n)$

$$\sigma h g(n) = \sigma h(n-2) = \sigma(j) \text{ for } j \in [n-2] \tag{51}$$
$$\sigma h g(n-1) = \sigma h(n-3) = \sigma(l) \text{ for } l \in [n-2], l \neq j \tag{52}$$

Then, we obtain

$$\frac{1}{|\mathbb{S}_{n-2}|} \sum_{h \in \mathbb{S}_{n-2}} f_i'(\sigma h g(n-1), \sigma h g(n)) = \frac{1}{(n-2)!} \sum_{j=1}^{n-2} \sum_{\substack{l=1 \\ l \neq j}}^{n-2} (n-4)! f'(\sigma(j), \sigma(l)) \tag{53}$$

$$= \frac{1}{n-2} \sum_{j=1}^{n-2} \frac{1}{n-3} \sum_{\substack{l=1 \\ l \neq j}}^{n-2} f'(\sigma(j), \sigma(l)). \tag{54}$$

So the term breaks in $\frac{1}{n-2}\sum_{j=1}^{n-2} P_i(g_7, \sigma, j)$. Here, $P_i(g_7, \sigma, j) = \frac{1}{n-3} \sum_{\substack{l=1 \\ l \neq j}}^{n-2} f'(\sigma(j), \sigma(l))$.

To further optimize the computation of $P_i(g_7, \sigma, j)$, we can compute partial sums $F_i(j) = \sum_{\substack{l=1 \\ j \neq i}}^{n} f_i'(j, l)$ and write

$$P_i(g_7, \sigma, j) = \frac{F_i(\sigma(j)) - f_i'(\sigma(j), \sigma(n-1)) - f_i'(\sigma(j), \sigma(n))}{n-3}. \tag{55}$$

Using the cyclic notation, we obtain $P_i(g_7, \sigma, j) = \dfrac{F_i(\sigma(j)) - \sum_{l \in \{3,6\}} P_i(g_l, \sigma, j)}{n-3}$ and $F_i(j) = \sum_{\substack{l=1 \\ j \neq i}}^{n} f_i((n-1, j, n, l))$.

$\square$

*Table 3.* Regression on QM9 and ZINC with different representations. We tested the following machine learning models: Extreme Gradient Boosting (Xgboost), Gradient Boosting Regressor (GBR), Elastic Net (EN), Linear Regression (Linear) using the default parameters of sk-learn. The error is measured as Mean Absolute Error.

| Dataset | Representation | Xgboost | GBR | EN | Linear |
|---------|---------------|---------|-----|-----|--------|
| QM9 | Single-orbit | 1.1560 | 1.1552 | 1.1571 | 1.1558 |
| | 3-orbits | 1.1570 | 1.1528 | 1.1559 | 1.1540 |
| | $C$'s eigs | 0.9602 | 0.9640 | 1.1571 | 0.9899 |
| | Laplacian's eigs | 0.9245 | 0.9377 | 1.0773 | 0.9704 |
| ZINC | Single-orbit | 1.5442 | 1.5432 | 1.5422 | 1.5423 |
| | 3-orbits | 1.5500 | 1.5441 | 1.5422 | 1.5423 |
| | $C$'s eigs | 1.3208 | 1.3876 | 1.5297 | 1.4249 |
| | Laplacian's eigs | 1.3350 | 1.3933 | 1.5353 | 1.4324 |

## D. Further experiments

In this section, we report further experiments carried on during the conference rebuttal period. These results are preliminary and intended to provide further insight, though they merit deeper future analysis.

### D.1. Scalability and other real-world datasets

To further evaluate the scalability and practical relevance of the multi-orbit $k$-Spectra, we extended the multi-orbit experiments on QM7 presented in Section 7.1 (Table 1) to two larger molecular datasets: QM9 and ZINC.

Both datasets were loaded through *torch.geometric.* QM9 consists of 130,831 molecules, each with up to 9 heavy atoms (maximum 29 nodes). We used the first 100,000 molecules for training and the remaining 30,831 for testing. ZINC contains 249,456 molecules with up to 38 nodes. We used the default train/test/validation split from `torch.geometric`, training on 220,011 molecules and testing on 5,000, ignoring the validation set for simplicity. Both tasks are regression problems. For QM9, the goal is to predict the dipole moment (target label 0 in `torch.geometric`); for ZINC, the task is to predict penalized logP, also referred to as constrained solubility.

Following the setup used for QM7, we compare different graph representations: the Single-Orbit Skew Spectrum on the off-diagonals, a 3-Orbit Skew Spectrum, and eigenvalues of adjacency and Laplacian matrices. For QM9, the three orbits are defined as follows: (1) the Coulomb matrix $C$; (2) an edge attribute obtained by summing the four edge attributes available in the dataset; and (3) the sixth node feature. For ZINC, the orbits correspond to: (1) the $G$ matrix; (2) the single edge attribute; and (3) the single node feature. We trained multiple regression models on these representations: Extreme Gradient Boosting (XGB), Gradient Boosting Regressor (GBR), Elastic Net (EN), Linear Regression (Linear) (Pedregosa et al., 2011).

Results are presented in Table 3. For QM9, , the 3-orbit representation consistently outperforms the single-orbit skew spectrum and is competitive with Laplacian and adjacency-based features, with the Laplacian eigenvalues achieving the best accuracy. In ZINC, the 3-orbit representation is again comparable to other methods, although the eigenvalues of the Coulomb matrix $C$ yield the best performance. We imagine that a more thoughtful use of the $k$-Spectra could yield better results.

### D.2. Integration with GNNs

To evaluate the practical impact of incorporating our graph representation into deep learning pipelines, we integrate the $k$-Spectra features into a standard Graph Neural Network. Specifically, we integrated our $k$-Spectra with the HGP-SL model (Hierarchical Graph Pooling with Structure Learning) (Zhang et al., 2019).

We follow a straightforward integration strategy: after computing a whole-graph embedding using our Doubly-Reduced $k$-Spectra, we concatenate this vector with the graph embedding produced by the final convolutional layer of HGP-SL. The concatenated representation is then passed to a fully connected layer that produces the final prediction. This design corresponds to the "post-concatenation" approach discussed in Section 5.

*Table 4.* Test accuracy on the PROTEINS dataset. Average $\pm$ standard deviation (Max) over six splits. $k$-Spectra features are concatenated to the HGP-SL embeddings, before the prediction layer.

| Baseline | **PR=0.5** $- 0.508 \pm 0.004$ (0.615) | | | **PR=0.2** $- 0.518 \pm 0.004$ (0.615) | | |
|---|---|---|---|---|---|---|
| | k=3 | k=4 | k=5 | k=3 | k=4 | k=5 |
| **Single-O** | $0.609 \pm 0.020$ (0.769) | $0.622 \pm 0.011$ (0.769) | $0.612 \pm 0.010$ (0.692) | $0.642 \pm 0.016$ (0.738) | $0.625 \pm 0.001$ (0.662) | $0.588 \pm 0.009$ (0.662) |
| **Multi-O** | $0.695 \pm 0.002$ (0.754) | $0.683 \pm 0.000$ (0.708) | $0.582 \pm 0.023$ (0.738) | $0.665 \pm 0.003$ (0.723) | $0.680 \pm 0.003$ (0.738) | $0.665 \pm 0.010$ (0.815) |

The model is trained and evaluated on the PROTEINS dataset, which contains protein structures represented as graphs, filtered to contain only graphs with 3 to 30 nodes. We use the same training protocol and hyperparameters as in the original HGP-SL work, including a learning rate of $0.001$, a batch size of $32$, and a maximum of $1000$ training epochs. Early stopping is applied based on validation loss. We experiment with two dropout rates: the original $0.5$ and a lower value of $0.2$.

We test both single-orbit and multi-orbit variants of DRkSP, using $k \in \{3, 4, 5\}$. For each configuration, we report the average accuracy, standard deviation, and maximum accuracy across six random train/test splits. Table 4 summarizes the results. The baseline HGP-SL model achieves an average accuracy of $0.508$ (dropout $0.5$) and $0.518$ (dropout $0.2$). Augmenting HGP-SL with the spectra consistently improves accuracy across all settings. Notably, moving from $k = 3$ to $k = 4$ generally improves performance. Multi-orbit variants, built using the adjacency matrix and one orbit per node feature, outperform single-orbit variants in most cases. These results support the hypothesis that $k$-Spectra features provide complementary information that enhances the learning capacity of GNNs.

