# OpenReview forum: "The Generalized Skew Spectrum of Graphs"
_ICML.cc/2025/Conference — ICML 2025 poster_

### Official Review · Reviewer_VVhY · 2025-03-14

**Overall Recommendation:** 4

**Summary:**

The paper generalizes the Skew Spectrum to obtain permutation invariant and isomorphism-invariant to graph embeddings. In detail, the authors propose multi-orbit spectra that can handle attributed graphs, multi-layer graphs and hypergraphs. Then $k$-Correlation Spectra is introduced to theoretically characterize the trade-off between complexity and expressivity, which is further extended to Doubly-Reduced $k$-Spectra. The theoretical results are supported by simulation and experiments.

**Claims And Evidence:**

The claims are well supported by evidence including theoretical analysis and experiments.

**Essential References Not Discussed:**

The paper lacks discussions on established work for graph invariants/expressivity [1, 2, 3]. It would also be better (but not necessary) to include some machine learning papers based on group theory [4, 5].


[1] Zhang, B., Zhao, L., & Maron, H. (2024). On the expressive power of spectral invariant graph neural networks. *arXiv preprint arXiv:2406.04336*.

[2] Gai, J., Du, Y., Zhang, B., Maron, H., & Wang, L. (2025). Homomorphism Expressivity of Spectral Invariant Graph Neural Networks. *arXiv preprint arXiv:2503.00485*.

[3] Zhou, C., Wang, X., & Zhang, M. (2023, July). From relational pooling to subgraph gnns: A universal framework for more expressive graph neural networks. In *International Conference on Machine Learning* (pp. 42742-42768). PMLR.

[4] Batatia, I., Geiger, M., Munoz, J., Smidt, T., Silberman, L., & Ortner, C. (2023). A general framework for equivariant neural networks on reductive Lie groups. *Advances in Neural Information Processing Systems*, *36*, 55260-55284.

[5] Dehmamy, N., Walters, R., Liu, Y., Wang, D., & Yu, R. (2021). Automatic symmetry discovery with lie algebra convolutional network. *Advances in Neural Information Processing Systems*, *34*, 2503-2515.

**Experimental Designs Or Analyses:**

See Methods And Evaluation Criteria part.

**Methods And Evaluation Criteria:**

The simulation part is convincing.

However, experiments on the real-world benchmarks can be strengthened. Particularly, the only real-world dataset is QM7, and more standard benchmarks including Zinc, QM9, ogbg-molhiv/molpcba, and Long Range Graph Benchmark, is strongly encouraged to validate the practical performance and scalability of the method. Moreover, the base models (e.g., XGB and Linear) do not include state-of-the-art GNNs, which is questionable. How to integrate (doubly-reduced) $k$-spectra into existing GNNs, and what is the effectiveness?

**Other Comments Or Suggestions:**

N/A

**Other Strengths And Weaknesses:**

This paper is valid in theory and is well written. It might be better for the authors to include more detailed introduction in group theory/graph invariants etc. to improve the readability for those readers not so familiar with this specific topic.

**Questions For Authors:**

* Can you connect the framework with some existing GNN expressivity framework, such as the WL hierarchy?

* Can you elaborate on more large-scale/real-world datasets and more state-of-the-art models?

**Relation To Broader Scientific Literature:**

The paper is related to literature in mathematics, including group theory, harmonic analysis, and graph isomorphism problem.

**Theoretical Claims:**

I check the correctness and did not find major flaws.

---

> ### Author Rebuttal · Authors · 2025-04-01
>
> **References:** Thank you for the references, we will include them in our manuscript.
>
> **Introduction:** More introductory material would help reach a broader audience, but the page limits presents a substantial challenge. Upon acceptance, we will consider both leveraging the extra page to revise the introductory material and extending the appendices to include a broader discussion on graph invariants, GNNs expressivity, and group theory.
>
> **WL hierarchy:** We thank the reviewer for the great suggestion. Relating our (general non-reduced) k-Spectra (kSP) to the WL hierarchy is very interesting, since understanding this relationship helps clarify the use of kSP in more advanced learning algorithms like GNNs. Preliminary analysis shows that kSP provides an alternative expressivity framework, since k-WL tests look at nodes and distinguish them based on having certain neighbors; kSP count different edge structures (see Intuition in Sec. 4). To investigate the relationship further, we extended our experiments on Doubly-reduced k-Spectra (DRkSP) to show that they are neither strictly less powerful than 1-WL nor strictly more powerful. Indeed, for undirected graphs of 7 nodes, we have that the concatenation of DRkSP for different values of k forms a complete invariant, while 1-WL tests, even with 50 iterations, report 22 collisions. On the other hand, on a dataset made of chordal graphs, 1-WL tests make 2 collisions and DRkSP 10. We updated our Figure 5 to report these results: https://anonfile.io/f/LRH87kqC
>
> **GNNs:** Most Message Passing Neural Networks (MPNN) architectures are at most as expressive as 1-WL tests [1, 2], while our previous experiment shows that DRkSP can be more powerful than 1-WL test. Corollary 3.14 of your second reference states that Spectral invariant GNNs can count cycles and paths up to 7 vertices, but non-reduced kSP, for high enough k, also counts paths and cycles on more than 7 vertices, see Intuition in Sec. 4. This suggests that DRkSP and further heuristics on kSP could be used to improve the expressivity of GNNs.
>
> We came up with three ways how to embed kSP into GNNs:
> - Precompute the whole-graph embedding with kSP and concatenate it to the GNN layer that outputs the graph embedding, just before the fully connected layer that computes the prediction.
> - Compute subgraph invariants in the node aggregation step using kSP.
> - Compute subgraph invariants in the pooling layer using kSP, instead of other invariants like sums or max of all node labels.
>
> In all the three cases, we could introduce some learnable parameters, such as weights in the direct sum terms of our Eq. 8 to tune the relevance of the orbits. We will include this discussion in the manuscript.
>
> To prove the point, we extend our experiments and report preliminary results. Following the first approach, we extend the HGP-SL architecture [3] and test on PROTEINS, using their code and filtering for graphs having between 3 and 30 vertices. We try a dropout rate of 0.5 and 0.2, use a learning rate of 0.001 and a batch size of 32, training the model for a maximum of 1000 epochs and using early stopping on validation loss. The results show that the DRkSP improves the learning capability and that both higher correlations and multiple orbits help: https://anonfile.io/f/yqPzRNyf
>
> **Real-world datasets and scalability:** To address the concern on scalability and practicality, in addition to our previous experiment on QM7, we added some preliminary experiments on two suggested real-world datasets: QM9 and ZINC. These two datasets contain 130,831 and 249,456 molecules, respectively. We processed these datasets on a desktop, using our non-optimized python implementation. This experiment extends Table 1 to the two datasets:https://anonfile.io/f/Y7HOKwqq
>
> For QM9 we used the first 100,000 molecules for training and the remaining 30,831 for testing. The 3-orbits include the adjacency matrix, the sum of the 4 edge attributes, and the 6th node feature. For ZINC, we used 220,011 molecules for training and 5000 for testing. The 3-orbits include the adj. matrix, the edge attributes, and the node features. The multi-orbit 3SP reports comparable results to the baselines, improving over the single-orbit on QM9. These experiments help prove the scalability of the approach. We are positive that a more thoughtful use of our kSP (including higher correlations, non-reduced versions, and further tuning) will lead to more competitive results.
>
> We remain available for further discussions/clarification.
>
> [1] Morris, Christopher, et al. "Weisfeiler and leman go neural: Higher-order graph neural networks." Proceedings of the AAAI conference on artificial intelligence. Vol. 33. No. 01. 2019.
> [2] Xu, Keyulu, et al. "How Powerful are Graph Neural Networks?." International Conference on Learning Representations.
> [3] Zhang, Zhen, et al. "Hierarchical graph pooling with structure learning." arXiv preprint arXiv:1911.05954 (2019).

---

> > ### Comment · Reviewer_VVhY · 2025-04-05
> >
> > I thank the authors for their rebuttal. The theoretical results that DRkSP is incomparable with 1-WL is interesting. The additional experiments are appreciated, and it would be better if the authors can try to integrate DRkSP with existing GNNs for all datasets in their camera-ready version. I have raised my scores.

---

> > > ### Author Response · Authors · 2025-04-09
> > >
> > > We thank the reviewer for their thoughtful comments and for recognizing the theoretical contribution regarding the incomparability of DRkSP with 1-WL. We appreciate their suggestion and are grateful for the improved scores.

---

### Official Review · Reviewer_JxLA · 2025-03-16

**Overall Recommendation:** 3

**Summary:**

This paper proposes a family of permutation-invariant graph embeddings, which generalizes the Skew Spectrum of graphs introduced by Kondor & Borgwardt (2008).  Grounded in group theory and harmonic analysis, the method introduces a new class of isomorphism-invariant graph invariants that can embed more complex graph structures, including attributed graphs, multilayer graphs, and hypergraphs, which the original Skew Spectrum could not accommodate.  The paper further defines a family of functions offering a tradeoff between computational complexity and expressivity, and demonstrates an improvement in expressiveness without additional computational cost through generalization-preserving heuristics.

**Claims And Evidence:**

yes

**Essential References Not Discussed:**

No

**Experimental Designs Or Analyses:**

yes

**Methods And Evaluation Criteria:**

yes

**Other Comments Or Suggestions:**

None

**Other Strengths And Weaknesses:**

1. The author claims that there is no increase in computational cost and has conducted an analysis on computation and complexity. However, there is no visualization comparing the complexity with existing algorithms.

2. The structure of the article is confusing. It spends a large amount of space introducing basic concepts, such as self-loops, node features, and edge features et al. However, the experiments are extremely insufficient, with almost no experimental results presented except for Figure 5.

**Questions For Authors:**

1. The author claims that there is no increase in computational cost and has conducted an analysis on computation and complexity. However, there is no visualization comparing the complexity with existing algorithms.

2. The structure of the article is confusing. It spends a large amount of space introducing basic concepts, such as self-loops, node features, and edge features et al. However, the experiments are extremely insufficient, with almost no experimental results presented except for Figure 5.

**Relation To Broader Scientific Literature:**

yes

**Theoretical Claims:**

yes

---

> ### Author Rebuttal · Authors · 2025-04-01
>
> **Computational cost:** Benchmarking the k-Spectra (kSP) and their Doubly-Reduced (DRkSP) version would be extremely interesting. However, a fair comparison requires a substantial amount of work, which has to be covered in future research. Fair benchmarking requires using the same programming language, amount of preprocessing, and hardware. While many existing algorithms leverage C implementations and GPUs or TPUs, our current implementation of kSP is in Python, deserves further optimization, and does not fully leverage the high parallelization opportunities of kSP, given by matrix sums, products and tensors. To perform the benchmarking, one would need to optimize for proper scaling and performance.
>
> While a full benchmarking is beyond the scope of this paper, our claim that the single-orbit DRkSP fully generalizes the Skew Spectrum (SkSP) without increasing its asymptotic computational cost is supported by an extremely detailed analysis and proof, which continues in Appendix C. The algorithm’s complexity is O(n^3). To get an idea of the scaling, this is the same as a full eigenvalue decomposition of an nxn adjacency matrix. More concretely, in the rebuttal period, even with a non-optimized implementation, we computed 3-orbit kSP of QM9 (130,831 molecules) and ZINC (249,456 molecules) on a desktop PC, confirming scalability.
>
> **Paper structure:** We appreciate the feedback on structure and we are sorry it can seem confusing. We structured our paper this way: Introduction + SkSP Background (3 pages), Multi-orbit + Higher correlations + heuristics (2.5), Computation and complexity (1 + Appendix), Experiments and Conclusions (1.5).
>
> We want to stress that from page 3 onwards, the article only presents our own contributions. Sec. 2 Background discusses the minimum group theoretic background to understand the original SkSP [1]. We believed that introducing these concepts was necessary for a broader audience, who might not necessarily be familiar with graph invariants, harmonic analysis on the symmetric group, and the Skew Spectrum. Indeed, Reviewer 3 (VVhY), suggests expanding the introductory material even further.
>
> About self-loops, node features, and edge features, we believe there is a misunderstanding. We assume that the reader is familiar with these concepts and never introduce them. The homonymous paragraphs in Section 3 describe, extremely concisely, how to encode these structures in our multi-orbit extension, which is part of the contribution.
>
> Our understanding is that the reviewer expected more pages for the experimental evaluation, as it is common in many ML papers. However, our structure is the one of a theoretical paper, with several formal statements, analysis, proofs, and experiments that showcase the theoretical results. We believe that similar papers are usually welcome at ICML. In any case, upon acceptance, we could expand the experimental section.
>
> **Experiments:** We acknowledge that the experiments can be improved (and we are currently working on this), but respectfully disagree on characterizing them extremely insufficient for our purpose. The current experiments confirm and illustrate our theoretical findings:
> - Eigenvalue collision shows the problem of multiple orbits and how simple eigenvalue invariants fail to address them.
> - Synthetic graphs shows how multi-orbit improves over the previous SKSP.
> - Table 1 shows that multi-orbits can offer more expressive representations than eigenvalue invariants and the SkSP on real-world datasets like QM7.
> - Figure 5 shows the expressive power of DRkSP, illustrating advantages over SkSP and graph spectral-eigenvalue methods.
>
> We further improved our experiments in response to feedback. In particular, we investigated the relationship between DRkSP and 1-WL tests, showing that DRkSP is not less powerful nor strictly more powerful than 1-WL (see https://anonfile.io/f/LRH87kqC). This suggests that kSP could improve the expressibility of standard GNNs, which are known to be less powerful than 1-WL [1, 2]. Indeed, a preliminary experiment on PROTEINS shows this is the case https://anonfile.io/f/yqPzRNyf
> Further experiments on QM9 and ZINC in the setting of Table 1 validate the scalability of the proposed method, with the multi-orbit representation offering comparable results to other representations https://anonfile.io/f/Y7HOKwqq
> More thoughtful experimental settings could show further advantage of multi-orbits on real-world datasets. Please check the replies to other reviewers for more information.
>
> We hope this rebuttal addresses the reviewer’s concerns and are happy to contribute with further information otherwise.
>
> [1] Morris, Christopher, et al. "Weisfeiler and leman go neural: Higher-order graph neural networks." Proceedings of the AAAI conference on artificial intelligence. Vol. 33. No. 01. 2019.
> [2] Xu, Keyulu, et al. "How Powerful are Graph Neural Networks?." International Conference on Learning Representations.

---

### Official Review · Reviewer_hv4C · 2025-03-18

**Overall Recommendation:** 3

**Summary:**

In this paper, authors extend the Skew Spectrum-based graph representation method to handle rich graph structures and achieve preferable computational efficiency. The idea is interesting.

**Claims And Evidence:**

A lot of theoretical analysis are made. The expression of the paper is easy to follow.

**Essential References Not Discussed:**

See summary.

**Experimental Designs Or Analyses:**

In this experiment, authors prove the effectiveness of the Multi-Orbit Embeddings, the Higher-Order Correlations and its advantage over the classic Skew Spectrum. More experimental results are encouraged to be conducted to illustrate the advantage of the proposed algorithm against the message-passing neural networks

**Methods And Evaluation Criteria:**

The setting is rational to me.

**Other Comments Or Suggestions:**

NA

**Other Strengths And Weaknesses:**

I have some concern over the advantage of giving each node in a graph an index. In this paper, authors claim that they have the advantage of “expressivity theoretical guarantees” and better generalization ability to richer graph structures like attributed graphs, multilayer graphs, and hypergraphs.

What can the theoretical guarantee be used for?

Also, there are many advanced algorithms that are proposed to handle attributed graphs, multilayer graphs, and hypergraphs, can you discuss more about the proposed method with those kinds of algorithms?

**Questions For Authors:**

NA

**Relation To Broader Scientific Literature:**

See summary.

**Theoretical Claims:**

I am not familiar with the theoretical analysis in the paper, so I am sorry that I am not able to appreciate the analysis in this part.

---

> ### Author Rebuttal · Authors · 2025-04-01
>
> **Message-Passing Neural Networks:** We stress that our main claim is not to improve message-passing neural networks (MPNN), but rather the Skew Spectrum of graphs. However, we think these methods can improve MPNNs and report evidence in the answers. Most MPNNs architectures aren’t more powerful Weisfeiler-Lehman (WL) tests, specifically 1-WL [1, 2]. We start from this and include two other experiments:
> - We investigate the relationship between k-Spectra (kSP) and WL tests. On a theoretical level, the two appear to be different: k-WL tests look at nodes and distinguish them based on having certain neighbors; k-correlations count different edge structures (see Intuition in Sec. 4). This suggests that kSP provides an alternative expressivity evaluation framework. We extended the experiments in Fig. 5 to show that Doubly-Reduced kSP (DRkSP) can be more and/or less expressive than 1-WL: DRkSP is complete on graph of 7 vertices, while 1-WL is not, but perform slightly worse on Chordal graphs. New Fig: https://anonfile.io/f/LRH87kqC
> - Since DRkSP can be more expressive than 1-WL it can be combined with GNNs to enhance their expressivity. We integrate DRkSP by concatenating the embedding with the GNN output, before a fully connected prediction layer. We use the HGP-SL model of [1], trained on PROTEIN with molecules between 3 and 30 nodes. The results show that the DRkSP improves the learning capability and that both higher correlations and multiple orbits help: https://anonfile.io/f/yqPzRNyf
>
> **Theoretical Guarantees:** We are unsure of what you mean by “giving each node of the graph an index”. We extend the Skew Spectrum in two ways: enabling more complex graph structures through multi-orbits; increasing the expressivity with higher correlations. An explanation of why the extensions improve expressivity is in both Section 3 and 4.
>
> The theoretical guarantees and the mathematical structure of the invariant guarantee interpretability. The invariant is not a black box. Having an explicit mathematical representation of the model allows determining why an algorithm made a specific choice, which is crucial, for instance, in cyberphysical and medical applications. The theoretical guarantees set expectations on the framework. For instance the embedding invariance (Theorem 4.3) implies that the invariant will not assign distinct representations to isomorphic graphs. Theorems 4.5 and 4.6 clarify which input adds information about graph isomorphism to the embedding. The intuitions in Sec. 3 and 4 explain how the kSP extends expressivity and help reasoning about WL tests and GNNs before experimenting. Other theorems detail the running time, guaranteeing a bounded execution time and setting expectations on the scalability of this method. If you consider it necessary, we can clarify this further in the revised manuscript.
>
> **Other algorithms:** Most of the modern approaches are either based on MPNNs [1] or on Spectral invariant GNNs [2]. We already discussed the relationship between MPNNs, WL and kSP above. About Spectral invariant GNNs, Corollary 3.14 of Reviewer 3’s second reference states that Spectral invariant GNNs can count cycles and paths up to 7 vertices, but non-reduced kSP, for high enough k, can count paths and cycles on more than 7 vertices, see Intuition in Sec. 4. This suggests that DRkSP and further heuristics on kSP could be used to improve the expressivity of such alternative methods.
>
> We can come up with three ways of integrating the kSP with GNNs:
> - Precompute the kSP and concatenate it to the GNN layer that outputs the graph embedding, just before the fully connected layer that computes the prediction (see our HGP-SL experiment).
> - Compute subgraph invariants in the GNN node aggregation step using kSP.
> - Compute subgraph invariants in the GNN pooling layer using kSP, instead of other invariants like sums or max of all node labels.
>
> In all the three cases, we could introduce some learnable parameters, such as weights in the direct sum terms of our Eq. 8 to tune the relevance of the orbits. We will include this discussion in the manuscript.
>
> We thank the reviewer for their kind feedback and hope that our answer addresses their concern. We remain available for further questions.
>
> [1] Morris, Christopher, et al. "Weisfeiler and leman go neural: Higher-order graph neural networks." Proceedings of the AAAI conference on artificial intelligence. Vol. 33. No. 01. 2019.
> [2] Xu, Keyulu, et al. "How Powerful are Graph Neural Networks?." International Conference on Learning Representations.
> [3] Zhang, Zhen, et al. "Hierarchical graph pooling with structure learning." arXiv preprint arXiv:1911.05954 (2019).
> [4] Lu, Q., Zhou, Z., & Wang, Q. (2024). Multi-layer graph attention neural networks for accurate drug-target interaction mapping. Scientific Reports, 14(1), 26119.
> [5] Feng, Yifan, et al. "Hypergraph neural networks." Proceedings of the AAAI conference on artificial intelligence. Vol. 33. No. 01. 2019.

---

### Decision · Program_Chairs · 2025-05-01

**Decision:**

Accept (poster)

**Comment:**

Generalization of the spectra to attributed graphs is a valuable contribution that can be utilized across the field; the theoretical ground is solid. Empirical results on 1WL expressivity as a results of the rebuttal to VVhY showcase the practical importance of the method. I suggest to accept the paper, it presents valuable new ideas to ICML, and is presented quite clearly.